# MatFormer: Nested Transformer for Elastic Inference

**Devvrit**[*△◇]   **Sneha Kudugunta**[*†◇]   **Aditya Kusupati**[*†◇+]

**Tim Dettmers**[†]   **Kaifeng Chen**[◇]   **Inderjit Dhillon**[◇△]   **Yulia Tsvetkov**[†]   **Hannaneh Hajishirzi**[†]

**Sham Kakade**[‡]   **Ali Farhadi**[†]   **Prateek Jain**[◇+]

[◇]Google DeepMind   [△]University of Texas at Austin   [†]University of Washington   [‡]Harvard University

## Abstract

Foundation models are applied in a broad spectrum of settings with different inference constraints, from massive multi-accelerator clusters to resource-constrained standalone mobile devices. However, the substantial costs associated with training these models often limit the number of unique model sizes that can be offered. Consequently, practitioners are compelled to select a model that may not be optimally aligned with their specific latency and cost requirements. We present MatFormer[2], a novel Transformer architecture designed to provide elastic inference across diverse deployment constraints. MatFormer achieves this by incorporating a nested Feed Forward Network (FFN) block structure within a standard Transformer model. During training, we optimize the parameters of multiple nested FFN blocks with varying sizes, enabling the extraction of hundreds of accurate smaller models without incurring additional computational costs. We empirically validate the efficacy of MatFormer across different model classes (decoders and encoders) and modalities (language and vision), demonstrating its potential for real-world deployment. We show that a 850M decoder-only MatFormer language model (MatLM) allows us to extract multiple smaller models spanning from 582M to 850M parameters, each exhibiting better validation loss and one-shot downstream evaluations than independently trained counterparts. Furthermore, we observe that smaller encoders extracted from a universal MatFormer-based ViT (MatViT) encoder preserve the metric-space structure for adaptive large-scale retrieval. Finally, we showcase that speculative decoding with the accurate and *consistent* submodels extracted from MatFormer can lead to significant reduction in inference latency. Project website.

## 1   Introduction

Large Foundation models [49, 45, 17] are deployed in a variety of settings with different compute and accuracy demands like real-time response on mobile phones or on multi-cluster GPUs for web-scale batch serving. However, typical model families provide only a few *independently trained* models of different sizes. For example, the Llama-2 family provides models with 7B, 13B, 34B, and 70B parameters [59]. So practitioners are forced to choose a smaller (and typically less accurate) model than their latency/cost budget. Alternatively, one can use compression/pruning to fit a bigger model in a given compute budget [19, 36, 53], but that requires additional training.

We introduce MatFormer, a natively elastic Transformer [61] architecture that overcomes this challenge. MatFormer allows for training one *universal* model which can be used to extract hundreds of smaller submodels without *any additional training cost* (Figure 1). MatFormer is a general

---

[*]Equal technical contribution. [+]Aditya Kusupati and Prateek Jain led the project.
Correspondence: devvrit@cs.utexas.edu,{snehakudugunta,kusupati,prajain}@google.com

[2]MatFormer stands for 🪆 **Mat**ryoshka Trans**former**, reflecting the model's inherent nested structure.

38th Conference on Neural Information Processing Systems (NeurIPS 2024).

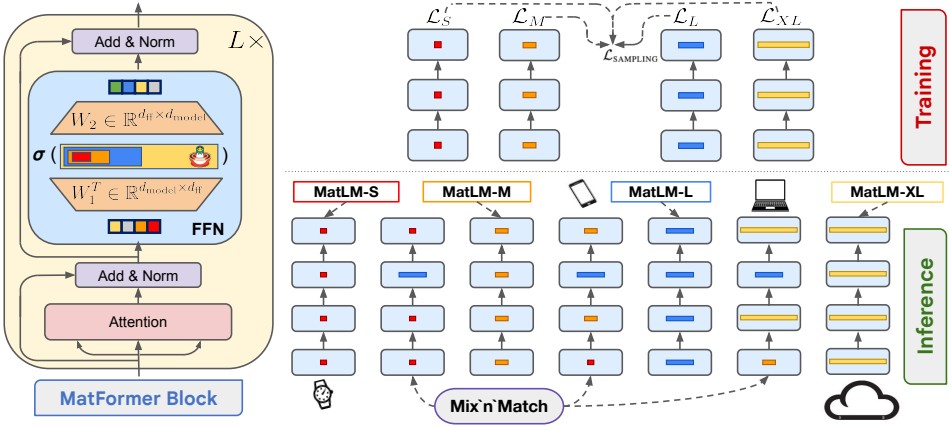

Figure 1: MatFormer introduces nested structure into the Transformer's FFN block & trains all the submodels, enabling free extraction of hundreds of accurate submodels for elastic inference.

architecture that can be applied to encoders and decoders, is domain agnostic, and is compatible with standard foundation model training pipelines.

MatFormer follows the principle of matryoshka representation learning [34], to introduce nested substructure inside the standard Transformer block. Formally, MatFormer defines Transformer blocks $T_i$, such that, $T_1 \subset T_2 \subset \cdots \subset T_g$, where $g$ is the number of nested transformer blocks, and $T_i \subset T_{i+1}$ relation indicates that the parameters of $T_i$ are contained in those of $T_{i+1}$. MatFormer can induce such sub-structure in both the attention and the feedforward network (FFN) blocks of the Transformer (see Figure 1). Consider, say, a FFN block that has $d_{\text{ff}}$ neurons in the hidden layer. Then, MatFormer induces matryoshka structure on these neurons, where $T_i$ contains the first $m_i$ neurons and $1 \le m_1 < m_2 \cdots < m_g = d_{\text{ff}}$ represent the number of neurons for each granularity or sub-model. Intuitively, this implies that the first $m_1$ neurons are "most significant" neurons as they belong to all the blocks followed by the next $m_2 - m_1$, and so on.

In a departure from related work (Section 2), despite optimizing for only $g$ granularities, we are able to extract exponentially more submodels post-training. Using the trained MatFormer blocks $T_1, \ldots, T_g$ at each layer, one can form new models by Mix'n'Match (Section 3.3), i.e., by taking an arbitrary combination of these blocks across layers. For example, in the first layer, one can select $T_g$, the largest block, choose $T_2$ in the second layer, and so on, forming $g^l$ different models (where $l$ is the number of layers). Surprisingly, in multiple settings, and for various model sizes, we observe that the extracted models indeed are accurate, with accuracy scaling with the size of the extracted model.

We train Matformer-based decoder-only Language Models (MatLM) up to 850M parameters and observe that: (a) MatLMs explicitly trained with $g$ exponentially spaced granularities outperform validation loss and one-shot downstream evals of respective $g$ baseline models trained independently from scratch, (b) our extracted models using Mix'n'Match lie on the accuracy-vs-parameters trade-off curve generated by the $g$ explicitly trained models, (c) through scaling experiments we observe that the loss vs compute law for different MatFormer models remains similar to vanilla Transformer models across different granularities and (d) the submodels extracted from MatLM have highly consistent behavior that is highly desirable for inference optimizations and deployment across scales.

We further study MatFormer-based ViT models (MatViT) and make similar observations. For example, MatViT-L/16 improves the accuracy of the standard ViT-L/16 model on ImageNet-1K, and the extracted sub-models all match or even perform better than the independently trained baselines. Furthermore, we demonstrate that, due to high consistency, MatViT models can be used as "elastic encoders" for adaptive image retrieval. That is, the metric-space of an image encoded by the universal (i.e. largest) MatViT model is roughly preserved by the nested submodels. Hence, based on query complexity, system load, and various other considerations, we can use one of the extracted MatViT encoders at inference time for retrieval on a fixed corpus encoded by the universal model – providing over $40\%$ less compute overhead with $< 0.5\%$ drop in accuracy.

**We make these key contributions:**

1. We introduce MatFormer, which incorporates a nested sub-structure within the standard Transformer and optimizes all the $g$ granularities to produce a single, universal elastic model.

2. We introduce Mix'n'Match, a simple heuristic with no computation overhead that finds optimal submodels within a given parameter budget, outperforming more complex NAS methods. This yields hundreds of accurate and consistent submodels without any training cost (Section 3).

3. MatFormer generalizes effectively to both decoder-only language models (MatLM) and vision encoders (MatViT), scaling as reliably and accurately as the standard Transformer, while enabling significantly faster autoregressive generation and large-scale adaptive dense retrieval (Section 4).

## 2   Related Work

Transformers [61] have become the unifying model architecture for foundation models [7] across modalities like language [8], vision [17] and audio [47]. While powerful, the standard Transformer block is not natively elastic in a way that enables large-scale adaptive and flexible deployment across various resource constraints. To cater to the plethora of deployment requirements, existing solutions include training a family of models of varying sizes [49, 2], post-hoc efficiency techniques like quantization [19], pruning [36], and distillation [53]. However, these solutions often are specific to the single constraint at hand, and require additional training for each new downstream usecase. This makes them far from being a truly elastic solution for adaptive deployment. Lastly, Transformer based LLMs are often sped-up during inference with techniques like speculative decoding [39, 12] – that benefits from the smaller draft & the larger verifier models having similar behavior – or early exiting [54] to enable real-time deployment.

Obtaining multiple smaller models from a single model has been explored in the past [66, 65, 9, 23, 10] with most work focusing on CNN encoders. In this work, we focus on Transformers in decoder-only language models and pretrained vision models. Specifically, OFA [9] trains a teacher CNN model, and uses distillation to finetune randomly sampled submodels (not nested) in the universal student CNN model. Moreover, OFA focuses on small scale models to be deployed on end devices. In contrast, MatFormer doesn't require distillation, thereby using substantially less memory, and uses nested models. Using nested models allows us to host multiple models together without significantly increasing the model's memory footprint, which is advantageous for scenarios where we want to route queries through different sub-networks. Slimmable networks [66] jointly optimizes and provide limited preset widths. Universal Slimmable network [65] extends this to sample from a continuous search space of submodels and optimizes them jointly. In contrast, MatFormer samples and only optimizes one of the preset granularities. HAT [63] trains a universal network only to learn relative performance for different architectures. For deployment, the authors use NAS to find the optimal architecture and *train it from scratch* before serving. In contrast, MatFormer requires no additional training and accurate subnetworks can be obtained using Mix'n'Match (Section 3.3) yielding results as good as NAS without the additional complexity. DynaBERT [27] jointly trains a fixed set of submodels, doesn't introduce any search strategy and discusses only using explicitly trained granularities as submodels. As a result of joint optimization of all granularities, DynaBERT yields fewer gradient updates and hence suboptimal performance compared to MatFormer while using same compute and memory (Section 4).

Moreover, we emphasize that while most work in this area optimizes for an *exponential number of models*, we optimize a small number of subnetworks ($g = 4$) to obtain an exponential number of models at inference time which leads to significantly better accuracy for large datasets. More

Table 1: Comparison of MatFormer with comparable techniques across training and inference. We emphasize that in contrast with earlier work, MatFormer requires optimizing fewer models to obtain an exponential number of models at inference time without the need for post-training or NAS. Moreover, MatFormer subnetworks are nested, allowing for adaptive retrieval & colocation of models during inference. Here, $l$ is the number of layers in the model and $\exp(l)$ refers to exponential in $l$.

| | N(Models Optimized) | N(Models Obtained) | Nested? | Model Selection | Post-training Needed? | Architecture | Decoder Model? |
|---|---|---|---|---|---|---|---|
| **MatFormer** | O(1) | $exp(l)$ | ✓ | Mix'n'Match | × | Transformer | ✓ |
| OFA [9] | $exp(l)$ | $exp(l)$ | × | NAS | × | CNN | × |
| Slimmable Networks [66] | O(1) | O(1) | ✓ | - | × | CNN | × |
| HAT [63] | $exp(l)$ | $exp(l)$ | × | NAS | ✓ | CNN | enc-dec |
| Sorted Network [60] | $exp(l)$ | $exp(l)$ | ✓ | - | × | Both | × |
| DynaBERT [27] | $O(1)$ | $O(1)$ | ✓ | - | × | Transformer | × |

recently, some of this work has been extended to Transformer encoders [11, 52] for extracting submodels in both static or dynamic settings. But they either fail at extending further to decoder-only language models ([52]) or perform suboptimal to MatFormer (Section 4) owing to differences in their training methodology. While not in the weight space, matryoshka representation learning [34] & FlexiViT [5] showcase elasticity in output & input spaces respectively by smoothly spanning deployment constraints with minimal overhead. MatFormer, in contrast, builds upon these works by creating nested structure within the weight space instead to enable truly elastic and adaptive Transformer-based (decoder & encoder) models that span all the accuracy-vs-compute tradeoff (statically or dynamically) with minimal changes and training overhead (Figure 1). SortedNet [60] is a concurrent work with similar goals which optimizes many sampled submodels (akin to prior work) unlike MatFormer's optimization of a few (typically 4) nested submodels. We also note FLEXTRON [64], a recent work that builds upon MatFormer by extending the nested elasticity in both MLP and Attention Heads simultaneously, tail patches the Matformer style training, and includes a router to automatically route each token within the different granularities in each layer.

In Table 1, we summarize the differences between MatFormer and the related work discussed. Among these works, we identified two central ideas commonly employed: jointly training multiple submodels and sampling random submodels. We consider DynaBERT [27] and Once-for-All (OFA) [9] as the respective most relevant prior works and compare them to MatFormer in Section 4.

# 3 MatFormer

In this section, we define MatFormer's nested substructure (Section 3.1) and discuss its training procedure for a chosen $g$ model granularities (Section 3.2). We then discuss elastic inference using Mix'n'Match models (Section 3.3) from MatFormer along with its deployment considerations.

## 3.1 MatFormer Structure

MatFormer defines $g$ Transformer blocks $T_i$, such that, $T_1 \subset T_2 \subset \cdots \subset T_g$ where $T_i \subset T_{i+1}$ indicates that the parameters of $T_i$ are contained in those of $T_{i+1}$. While it is possible to impose such a structure on any part of the Transformer, we select the FFN block to define our method and present a majority of our experiments, as the model size and computational cost of a Transformer is dominated (around $60\%$ for LLMs and ViTs) by the FFN block (see Appendix C, and Appendix F.2 for experiments applying MatFormer to the attention block of the Transformer). So, in this work, we focus on inducing the MatFormer's nested sub-structure in the FFN block. We then stack individual blocks (for $l$ layers) to form $g$ nested models ($\mathcal{M}_{1\cdots g}$) with shared parameters i.e., $\mathcal{M}_i \subset \mathcal{M}_{i+1}$.

The Transformer FFN block has a single hidden layer with $d_{\text{ff}}$ neurons and both input and outputs in $\mathbb{R}^{d_{\text{model}}}$, and fixed FFN ratio $:= d_{\text{ff}}/d_{\text{model}}$ (typically $\geq 4$). MatFormer introduces the matryoshka nested structure with $g$ granularities on the hidden representation of the FFN block. Concretely, a nested sub-block of the Transformer, $T_i$ contains the first $m_i$ neurons of the FFN and $1 \leq m_1 < \cdots < m_g = d_{\text{ff}}$ represent the number of neurons for each granularity or sub-model. So, depending on the chosen granularity the FFN operation of $T_i$ i.e., $T_i^{\text{FFN}}$ on an input $x \in \mathbb{R}^{d_{\text{model}}}$ is:

$$T_i^{\text{FFN}}(x) = \sigma(x \cdot \mathbf{W}_1[0:m_i]^\top) \cdot \mathbf{W}_2[0:m_i], \tag{1}$$

where the weight matrices of FFN are $\mathbf{W}_1, \mathbf{W}_2 \in \mathbb{R}^{d_{\text{ff}} \times d_{\text{model}}}$ and bias terms are omitted for simplicity. $\mathbf{W}_1[0:k]$ denotes the submatrix with the first $k$ rows of $\mathbf{W}_1$. Finally, $\sigma$ is a non-linearity often set to GELU [24] or squared ReLU [56]. In this work, we chose the $g = 4$ exponentially spaced granularities with FFN ratios of $\{0.5, 1, 2, 4\}$ i.e., the nested hidden neurons are of the sizes $\{\frac{d_{ff}}{8}, \frac{d_{ff}}{4}, \frac{d_{ff}}{2}, d_{ff}\}$. We get $g$ nested submodels $\mathcal{M}_1 \subset \mathcal{M}_2 \ldots, \subset \mathcal{M}_g$ where $\mathcal{M}_i \leftarrow [T_i]^{\times l}$, i.e., $\mathcal{M}_i$ is formed by stacking $T_i$ for $l$ layers. The input and output embedding matrices are shared across the models.

We note that we can form a similar sub-structure on the attention heads, with the heads being organized from "most" to "least" significant, where the more significant heads are shared by more sub-models. That is, we use the first $m_i$ attention heads for the $i$th granularity. We can also introduce this sub-structure in the token embedding ($d_{\text{model}}$) supplied to each Transformer block.

## 3.2 Training

For a Transformer model $\mathcal{M}$, the forward pass on an input $x$ is denoted by $\mathcal{M}(x)$ and let $\mathcal{L}$ denote the loss function between the output and the target $y$: $\mathcal{L}(\mathcal{M}(x), y)$.

MatFormer relies on a simple training strategy of randomly sampling the $g$ nested submodels across training. To this end, for each step we randomly sample a Matformer granularity $i = 1, 2..., g$ and train for it using the standard stochastic gradient-based optimizers [55]:

$$\mathcal{L}_{\text{SAMPLING}}(x, y) = \mathcal{L}(\mathcal{M}_i(x), y), \tag{2}$$

where $\mathcal{M}_i$ is the parameter set of $i$-th granular submodel, with $\mathcal{M}_i$ chosen from a probability distribution $\{p_1, p_2...p_g\}$. For most experiments in this paper, we uniformly sample each submodel - in Appendix F.3, we find that tuning this probability distribution can result in stronger submodels.

MatFormer training results in $g$ accurate nested submodels $\mathcal{M}_{1...g}$ inside the universal MatFormer model ($\mathcal{M}_g$), and also enables the extraction of hundreds of smaller submodels along the accuracy-vs-compute curve traced by the $g$ explicitly optimized submodels (Section 3.3). These models emerge for free using Mix'n'Match during inference and drastically reduce the amortized training cost per model obtained through MatFormer. This method results in smaller submodels that have highly consistent behavior (Section 3.4) with the universal model.

## 3.3 Mix'n'Match

At inference time, it is trivial to extract one of the $g$ submodels $\mathcal{M}_1 \subset \mathcal{M}_2 \ldots, \subset \mathcal{M}_g$ by stacking the corresponding Transformer block $T_i$ across layers. However, by selecting different granularities for each MatFormer layer, it is possible to generate a combinatorially large number of accurate smaller models for free. We call this simple procedure *Mix'n'Match* and observe that these additional model granularities – which were never explicitly optimized – are highly performant.

For a given compute or parameter budget, there are multiple possible submodels. A common strategy to select an optimal submodel is Neural Architecture Search (NAS) [48, 68]. This, however, is computationally expensive (Appendix D.2). With, Mix'n'Match, we propose gradually increasing the sub-block size with the "least slope". More concretely, we recommend selecting sub-blocks with minimal granularity changes across layers, ensuring that the size of the $j^{th}$ layer is at least that of the $i^{th}$ layer for $j > i$. To give a concrete example, we find that a submodel that uses granularity $g_2$ for half the layers then $g_3$ for the rest will likely be better than a submodel that uses $g_1$ and $g_4$ for a similar model size. Our heuristic is underpinned by the training methodology, where each sampled subnetwork maintains consistent layer granularity across the model. Consequently, the model adapts best to configurations where layer granularities are either uniform or display minimal variation. This intuition is also backed by NAS, which predicts balanced configurations over skewed (Appendix D.1). Among these "balanced" configurations, we empirically found the increasing with minimum slope configuration to perform the best. In Section 4.1.1 and Appendix D.1, we show that Mix'n'Match works at least as well as using evolutionary search based NAS methods [48] as used by OFA [9].

To summarize, we find that using Mix'n'Match is a simple, cheap and effective hueristic to select a highly performant submodel for a given compute budget (Sections 4.1.1 & 4.2). We provide further details and intuition in Appendix D.1.

## 3.4 Deployment

The design of MatFormer is beneficial for both *static and dynamic workloads*:

**Static Workloads** To expand upon the example of Llama-2 models in Section 1, a deployment setup might, say, have the latency budget to support 40B parameter Llama model, but can only host a 34B variant because the next bigger model (70B) has significantly higher latency. Training a 40B parameter from scratch would require $4.8 * 10^{23}$ FLOPs, when training a 34B model and 70B model already cost $4.08 * 10^{23}$ and $8.4 * 10^{23}$ FLOPs respectively. So, one would need to settle for a less accurate model despite the larger latency budget. With Matformer, one could obtain a highly accurate 40B model for 0 additional training FLOPS. More precisely, for static workloads, where compute resources are known beforehand and the inputs remain relatively similar in difficulty, one can choose the most accurate static submodel for the constraints using Mix'n'Match.

**Dynamic Workloads** For dynamic workloads, where the compute resources or the input hardness change on the fly, we can use the universal MatFormer model to dynamically extract the optimal submodel for each token or query. This works especially well for MatFormer because all the extracted submodels have high behavioral *consistency* with universal MatFormer model (Section 4.1) – minimizing the drift across predictions from various submodels. We measure the consistency between two generative models as the *percentage of matching tokens* generated by them for the same prefix or using the *KL divergence* of the smaller model outputs with the larger model outputs – this accounts for potential sampling strategies in decoding. This high consistency results in superior inference time speedups for techniques like speculative decoding [39] (Section 4.1.1) and can assist in reducing prediction drift between cross platform deployments. We also show that higher model consistency also aids metric-space structure preservation in encoder models (Section 4.2.2). Moreover, given the nested architecture of MatFormer, model colocation can be more memory efficient.

## 4 Experiments

In this section, we empirically evaluate MatFormer across modalities (language in Section 4.1 and vision in Section 4.2) and model classes (decoder and encoder). We demonstrate the elastic deployment of MatFormer-based models (Sections 4.1.1 & 4.2) for tasks spanning from one-shot generative evals to adaptive image retrieval. Additionally, we also investigate the reliable scaling behavior [29] of MatFormer models (Section 4.1.2).

### 4.1 MatLM: MatFormer Language Models

**Experiment Setting:** We build MatFormer-based decoder-only Language Models – MatLMs – and contrast them to their vanilla Transformer counterparts (LMs) [41]. For each MatLM model with fixed $d_{model}$, we optimize for $g = 4$ nested granularities represented by FFN ratios of $\{0.5, 1, 2, 4\}$ – i.e., only the hidden representation size of the FFN block changes. We denote these submodels as MatLM – $\{S, M, L, XL\}$ in increasing order of model size and refer to MatLM-XL as the universal MatLM. For baselines, we train vanilla Transformer models with comparable architectures. That is, for each MatLM, we train 4 separate baseline models with FFN ratios of $\{0.5, 1, 2, 4\}$ denoted as Baseline – $\{S, M, L, XL\}$. In addition, we adapt OFA [9] and DynaBERT [27] to our language modeling setup, and compare those to MatFormer at the same model size. We evaluate these models on validation loss and average accuracy on 25 English tasks [8, 22, 3]. We note that no additional memory and compute is used during training these methods compared to independently trained Baselines. Please see Appendix B for further details on training, baselines, evaluation, and the datasets.

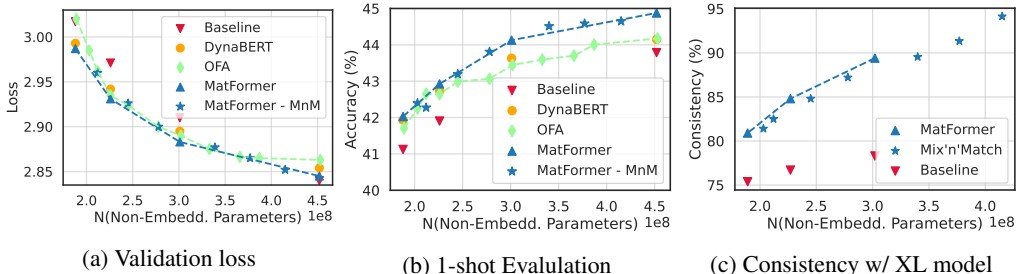

(a) Validation loss  (b) 1-shot Evaluation  (c) Consistency w/ XL model

Figure 2: Validation loss & one-shot downstream evaluation scores for the 850M MatLM & baseline models. Mix'n'Match helps generate accurate and more consistent models from MatLM that lie on the performance-vs-compute curve spanned by the explicitly optimized submodels.

**Results compared to baselines:** To showcase efficacy of MatFormer over baselines, we evaluate 850M MatLM model with the corresponding baseline counterparts in Figure 2.

Overall, in Figures 2a and 2b we observe all granularity submodels of MatLM outperform their baseline counterparts. Specifically, we find that DynaBERT exhibits a significant 0.01 log perplexity gap compared to MatFormer on the 850M model. The underlying reason is DynaBERT employs joint optimization of all granularities, which yields to fewer gradient updates and hence suboptimal performance compared to MatFormer. DynaBERT would require more than 15% extra compute to perform as well as MatLM. OFA, similar to MatFormer, maintains a single universal model but

employs random subnetwork sampling during its training. This leads to sampling fewer models close to S and XL granularity, resulting in inferior performance in this regime. The performance gap manifests as a bell-shaped loss curve (Figure 2a), highlighting OFA shortcomings in handling the trade-off between maintaining universal (XL) model quality and model elasticity. Additionally, OFA's training necessitates complicated NAS strategy for optimal submodel selection. However, using NAS at scale is costly and erroneous, which we further discuss in Appendix D.2. We refer the reader to Appendix B.4 for a more detailed discussion of MatFormer performance compared to baselines, and advantages and downsides of each method.

### 4.1.1 Elastic Inference with MatLM

**Accurate MatLM submodels for every constraint for free with Mix'n'Match.** Mix'n'Match enables the MatLM to deliver accurate models for any compute constraint between S and XL, beyond the fixed granularities {S, M, L, XL}. We assess the efficacy of Mix'n'Match on the 850M parameter MatLM, comparing validation loss and downstream performance against independently trained baseline models {S, M, L, XL}. Figure 2a demonstrates that Mix'n'Match achieves optimal loss-vs-compute trade-offs at no additional cost. Additionally, downstream evaluations in Figure 2b reinforce this trend. In deployment scenarios with only 55% of the compute resources for a MatLM-XL model, a Mix'n'Match submodel approximates the XL's performance with only about a 1% accuracy drop, compared to a 2% drop when using the MatLM-M model. This highlights the efficiency of Mix'n'Match in creating numerous optimal models, as exemplified by selected instances along the performance curves.

We experimented with several heuristics to select the best subnetwork, but consistently observed that gradually using larger granularities in deeper layers worked the best (Section 3.3). We find that this heuristic better than the evolutionary search based techniques [48], used in OFA [9] in Figures 2a & 2b. We also find that applying NAS to MatFormer provides no benefit over Mix'n'Match in Figure 6. We discuss additional details on Mix'n'Match in Appendix D.1.

**MatLM submodels speed up speculative decoding.** Speculative decoding leverages an accurate lightweight LM as a draft model to autoregressively generate a few tokens, followed by verifying these drafts with a larger model through parallel decoding on the generated tokens. When the draft is inaccurate, the draft model is rolled back and reset to the larger model's output. This results in considerable inference speed-up for the *same accuracy as the large model*. We point the reader to the original paper for a more detailed explanation [39].

Slow down of this algorithm stems from cases where the smaller model's predictions disagree with the larger model. A draft model that is significantly more consistent with the larger verifier model would lead to less rollbacks of the draft predictions and therefore lower latency. As seen in Figure 2c, MatLM submodels can be up to $11.5\%$ more consistent than the baselines to their corresponding XL model. The significant gap persists even in the KL divergence variant of consistency with the XL model's outputs (see Figure 7 in Appendix). This improved consistency along with the need for only a single universal model positions MatLM favorably to improve techniques that require draft and verifier models such as speculative decoding.

Table 2 shows the inference time speed-ups from speculative decoding using the S and XL submodels of the 850M language model for drafting and verification respectively. Speculative decoding with independently trained baseline LMs results in a speed-up of up to $10\%$ over the standard autoregressive decoding of the 850M-XL model. But MatLM-based speculative decoding is up to $6\%$ faster than traditional speculative decoding. This additional speed-up can be primarily attributed to the more consistent nature

Table 2: Inference time speed-ups over a standard 850M model through speculative decoding using a 393M (S) draft and 850M (XL) verifier model.

| Speculative Decoding | LAMBADA | TriviaQA |
|---|---|---|
| Baseline | 1.10× | 1.08× |
| MatLM | 1.14× | 1.11× |
| + shared attention cache | 1.16× | 1.14× |

of MatLM-based drafter and verifier models and is further boosted by the ability to share attention cache across models from MatLM which is infeasible for the baselines (see Appendix C.1). Finally, MatLM further reduces the memory overhead for inference by removing the need to have two models during resource-constrained deployment.

### 4.1.2 MatLM Scales as well as Vanilla Transformer LMs

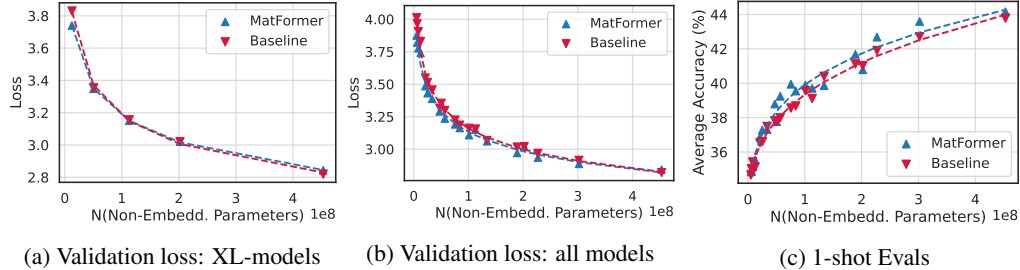

(a) Validation loss: XL-models     (b) Validation loss: all models     (c) 1-shot Evals

Figure 3: We train various decoder-only MatLM models at a range of sizes from 78M to 850M parameters and observe the scaling trends of all granularities (S, M, L, XL) for validation loss and 1-shot downstream evaluation scores. We find that the MatLM-XL models across scales mimic the training trends of Baseline-XL models. Interestingly, we also note that that validation loss and downstream evaluations follow the *scaling trends of the XL-models across all granularities*.

Now that we have established that a 850M MatLM model and its submodels are at least as accurate as the baseline Transformer LMs, we want to examine the scalability of training MatLM models. So, we study the scaling properties [29, 25] of MatLMs and compare them to vanilla Transformer baseline LMs trained for the same number of tokens. We train models ranging from 78M to 850M parameters on 10B to 80B tokens (per granularity) and plot the validation loss for MatLM – {S, M, L, XL} compared against independently trained baselines in Figure 9.

First, in Figure 3a, we observe that the training of MatLM-XL models across model sizes scale as reliably as the Baseline-XL LMs for loss vs. number of parameters. Figure 3b interestingly shows that all granularities {S, M, L, XL}, of MatLM and Baseline follow the same scaling trend. Therefore, we fit a scaling law according to the number of non-embedding parameters ($N$) and training tokens ($D$) for all possible submodels for both MatLMs and the baselines in

Table 3: Fitted parameters for the scaling equation: $\text{Loss}(N, D) = a \cdot (ND)^b + c$

|           | a     | b     | c    |
|-----------|-------|-------|------|
| Baseline  | 14.08 | -0.10 | 0.89 |
| Matformer | 21.60 | -0.13 | 1.33 |

Table 3. We observe that the fitted parameters are extremely similar, suggesting that MatLMs scale similarly to vanilla Transformer LMs. In Figure 3c we also find that the downstream evals for MatLM 0.3% better than the baselines, with the smaller submodels further outperforming the baselines at scale by upto 1.4%. Finally, Figure 9f in the Appendix shows that the MatLM submodels are more consistent with their XL model compared to the baseline counterparts across scales.

We note that the scaling laws do not capture how MatLMs have been optimized for multiple submodels and even have performant submodels that have not been explicitly optimized for (Section 4.1.1) We leave formulations that capture these subtleties to future work and further discuss this in Appendix E.1. We provide full results split by granularity in Appendix E.

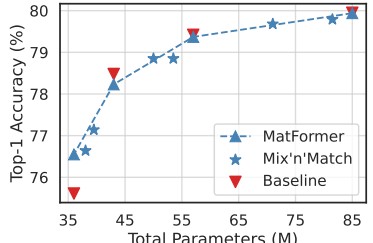

(a) B/16 trained on ImageNet-1K with AugReg     (b) L/16 pretrained on IN-21K $\rightarrow$ ImageNet-1K.

Figure 4: MatViT variants match or outperform standard ViT models on ImageNet-1K classification and provide free extracted models that span the accuracy-compute curve through Mix'n'Match.

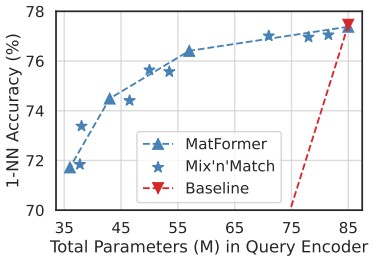
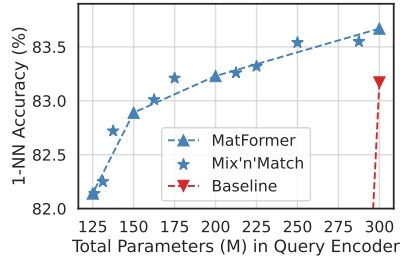

| (a) B/16 trained on ImageNet-1K with AugReg | (b) L/16 pretrained on IN-21K → ImageNet-1K. |

Figure 5: MatViT natively enables elastic encoders for adaptive retrieval that can be used for real-time query side computation while retaining strong accuracy on ImageNet-1K, unlike the baselines.

## 4.2 MatViT: MatFormer Vision Transformers

We extend MatFormer to Vision Transformer (ViT) [21] based computer vision encoder models. MatFormer-based ViT (MatViT) enables elastic inference for fundamental tasks like image classification and retrieval. We train the MatFormer variant of the standard ViT-B/16 and ViT-L/16 models – MatViT-B/16 and MatViT-L/16 that are trained with $g = 4$ nested granularities (FFN ratios of $\{0.5, 1, 2, 4\}$). B/16 models are trained on ImageNet-1K [50] with AugReg [57] while L/16 models are pretrained on ImageNet-21K [18] followed by finetuning on ImageNet-1K. All models use the training setup and optimal hyperparameters of standard ViT variants from the Scenic library [16].

### 4.2.1 Image Classification

For image classification, we evaluate both ViT & MatViT models on ImageNet-1K. Figure 4a shows that the explicitly optimized granularities in MatViT result in as accurate models as the independently trained baselines for the B/16. However for L/16, as shown in Figure 4b, we see that the MatViT models are up to $0.35\%$ more accurate than the baseline for the same inference cost.

We then explore using MatFormer at different training stages with a $2\times2$ grid of pretraining-finetuning pairs (Table 7 in Appendix G.1) and find that using a MatFormer during pretraining helps bring more accurate and flexible encoders for downstream use. Further, finetuning using MatFormer enhances elastic deployment depending on the constraints at hand through Mix'n'Match.

**Adaptive Encoders with Mix'n'Match.** Furthermore, our Mix'n'match models' accuracy almost lies on the line joining accuracy of explicitly trained granularities. In scenarios where, say, an application can host 50M parameter B/16 model, MatViT can provide $0.8\%$ more accurate model than the current approach which would host the largest baseline model with $\leq$ 50M parameters.

During deployment, the universal MatViT model can be stored in memory and depending on the compute constraints be used to extract an adaptable smaller model to maximize accuracy with the available resources at that moment. Currently, we find the Mix'n'Match models on the accuracy-compute curve through a quick inference on the validation set. While relatively scalable, this points to the need for optimal budget allocation across layers in neural networks [33].

### 4.2.2 Adaptive Image Retrieval

The goal of image retrieval is to find semantically similar images – e.g. images from the same class – using representations obtained from a pretrained encoder [13]. Standard approach is to encode the database images as well as query image with same encoder and run nearest neighbor retrieval for the query embedding. While we can embed database images with an expensive encoder, the query encoder generally has to be real-time. Furthermore, the setting of query encoding might be varied, e.g., on-device vs. cloud processing, varying query load and query complexity. Current solutions have a fixed encoder thus compromising on accuracy or cost for various settings.

Given the elastic nature of MatViT, it is a good candidate for query encoder. However, retrieval also requires that submodels preserve distances between fixed database (with large encoder) and query embeddings across all the granularities. If we use smaller baseline ViT models only for query encoding, these distances are not preserved and lead to nearly 0 retrieval accuracy (see Figure 5).

We evaluate both ViT and MatViT encoders on ImageNet-1K for image retrieval. We compute 1-nearest neighbor (NN) accuracy using the representation vector of the [CLS] token (also see Appendix G.2). Figure 5 shows that submodels extracted from MatViT can approximately preserve distances and provide significantly more flexibility. For example, with a loss of $< 0.5\%$ accuracy, MatViT-L/16 can reduce compute cost by $40\%$. This corresponds to the 175M Mix'n'Match parameter model in Fig 5b, which is 40% smaller than the 300M XL model, and has $< 0.5\%$ accuracy drop. To our knowledge, this is the *first result of its kind* and opens up a wide variety of adaptive inference strategies for large-scale semantic search.

## 5   Conclusion

In this work we presented MatFormer, a natively elastic Transformer architecture that allows training a single universal model which can be used to extract hundreds of smaller accurate submodels at zero additional cost at deployment time. We find that the MatFormer Language Model (MatLM) matches the perplexity & 1-shot accuracy of independently trained models. In fact, MatLM demonstrates an interesting loss-vs-compute scaling curve that is nearly *independent* of trained granularity indicating robust generalization to *extremely* large models as well. Finally, MatFormer submodels enable diverse inference time speedups like faster autoregressive generation with speculative decoding and elastic query encoders for adaptive dense retrieval across modalities. We believe dynamically routing these models to change inference latency [32, 40, 20], and developing the hardware optimizations required is a promising area for future work.

## Acknowledgments

We are grateful to Aishwarya P S, Yashas B.L. Samaga, Varun Yerram, Lovish Madaan, Anurag Arnab for assistance in setting up training pipelines, Matthew Wallingford, Praneeth Netrapalli, Orhan Firat, Rohan Anil, Tom Duerig, Luke Zettlemoyer, Manish Gupta, Rahul Sukthankar and Jeff Dean for helpful discussions, support and feedback.

We also acknowledge the computing resources and support from HYAK at the University of Washington, FAS RC at Harvard University, Kempner Institute and a GCP credit award for the early-stage exploration of this project. Ali Farhadi acknowledges funding from the NSF awards IIS 1652052, IIS 1703166, DARPA N66001-19-2-4031, DARPA W911NF-15-1-0543, and gifts from Allen Institute for Artificial Intelligence and Google. Sham Kakade acknowledges funding from the Office of Naval Research under award N00014-22-1-2377. This work has been made possible in part by a gift from the Chan Zuckerberg Initiative Foundation to establish the Kempner Institute for the Study of Natural and Artificial Intelligence. Yulia Tsvetkov acknowledges support from the National Science Foundation under CAREER Grant No. IIS2142739, NSF grants No. IIS2125201 and IIS2203097, and gift funding from Google, MSR, and OpenAI. Hannaneh Hajishirzi acknowledges funding through a gift from Allen Institute for Artificial Intelligence.

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

# A   Broader Impact Statement

The elasticity of MatFormer enables its use in diverse deployment scenarios. This lowers the barrier for practitioners to use foundation models tailored to their deployment scenarios. Training foundation models remains expensive, with the largest models we discuss trained on 256 TPU-v4 cores for 3 days. Moreover, while we report validation loss and downstream evaluation scores on a variety of tasks, we acknowledge the possibility that MatFormer can have adverse effects on bias [26] or underrepresented domains/languages [1].

# B   Implementation Details

## B.1   Architecture and Training

For our experiments, we train a range of MatLMs varying from size 78M to 850M for 10B-80B tokens – we scale model size equally with the number of training tokens [25]. For each MatLM granularity, we also train a corresponding baseline vanilla Transformer model. That is, for each model size we train Baseline-XL, L, M, S with $d_{ff} = 4 * d_{model}, 2 * d_{model}, d_{model}, d_{model}/2$. All models have 16 layers, 16 attention heads, and a $d_{model} : d_{ff}$ ratio of $1 : 4$. We train a 256k vocabulary using the SentencePiece library [31], use a maximum context length of 1024 tokens, and a batch size of 1M tokens. We pretrained the 850M models on 256 v3 TPU chips. We provide further details on these models in Table 4. For further details on training setup, we point the reader to [58].

Table 4: Model details for the models scales used to conduct the experiments described in Section 4.1, with a breakdown of total parameter counts, non-embedding parameter counts and FFN parameter counts for each model granularity.

| Parameter Count (full / spliced) | Non-Embedding Params (full / spliced) | FFN Params (full) | $d_{model}$ | N(tokens) |
|---|---|---|---|---|
| 78M (74M / 72M / 71M) | 12.6M (8.4M/6.3M/ 5.3M) | 8.4M | 256 | 10B |
| 180M (164M / 157M / 152M) | 50M (33.7M/25.3M/21.1M) | 33.6M | 512 | 20B |
| 310M (272M / 253M / 244M) | 113M (75M/56M/47M) | 75.6M | 768 | 30B |
| 463M (397M / 363M / 346M) | 201M (134M/100M/84M) | 134M | 1024 | 40B |
| 850M (696M / 620M / 582M) | 453M (302M/227M/189M) | 302M | 1536 | 80B |

## B.2   Downstream Evaluation

We evaluate all the LM models trained on set of 25 English tasks similar to [8, 22, 14, 3], including:

1. **Open-Domain Closed-Book Question Answering tasks**: TriviaQA [28], Natural Questions [35], and WebQuestions [4].

2. **Cloze and completion tasks:** LAMBADA [46], HellaSwag [67], and StoryCloze [43].

3. **Winograd-style tasks:** Winograd [38] and WinoGrande [51].

4. **Reading comprehension:** RACE [37].

5. **Common sense reasoning:** PIQA [6], ARC [15], and OpenBookQA [42].

6. **SuperGLUE** [62]

7. **Natural language inference:** Adversarial NLI [44].

For all the granularities corresponding to each model, we present evaluation numbers along with development set log perplexity loss on all the 25 tasks in Tables 9 to 13.

## B.3   Baseline Implementation Details

**Independently trained Baseline:** Baselines are trained from scratch independently, where each granularity in a given model size uses X tokens mentioned in Table 4. For example, for 850M model X = 80B tokens. Therefore, Baseline-{S, M, L, XL} process 4X tokens in total. Baselines trained from scratch result in the same number of models that are explicitly trained for. That is, there is no way to directly get a submodel of size in between the trained granularities without using additional methods like distillation, pruning, etc.

**OFA:** OFA [9] trains a big model from scratch, then freezes it and starts another run of a similar sized "universal model" where it samples random subnetworks and optimizes them with distillation and ground truth loss. The central idea of this work is sampling random submodels and optimizing them. Distillation is a orthogonal component which can be added to MatFormer as well. In order to use comparable compute and memory to the Baseline (Section 4.1), we modify the method to only optimize w.r.t. ground truth loss - as also done in the case of MatFormer. Overall, OFA uses 4X tokens, same as all the baseline granularities trained form scratch. In each step, OFA samples a random model, where each of its layer has {S, M, L, XL} granularity sampled randomly, and optimizes it.

To obtain submodels according to a given cost constraint, OFA proposes using NAS [48] to search for optimal architecture. Specifically, OFA samples a random set of (architecture, loss) pairs from the universal model. Then, it trains a predictor on this pair, taking architecture as input and predictiing the loss. Finally, it uses an evolutionary search based routine [48] to search for the optimal architecture within a specified constraint, using the predictor in its search routine. More details of NAS and comparison with MatFormer Mix'n'Match is presented in Section D.

**DynaBERT**: DynaBERT [27] has $g$ fixed granularities of submodels - similar to MatFormer. Similar to OFA, DynaBERT employs distillation loss as well. For a comparable analysis and to maintain compute/memory comparable to baseline, we only optimize DynaBERT wrt ground truth loss, similar to MatFormer.

The crucial difference between DynaBERT and MatFormer lies in its training methodology. In each step, DynaBERT takes 4 batches of data, optimizes each batch with each of the {S, M, L, XL} granularity and averages these losses. Similar to Matformer and OFA, it processes a total of 4X tokens. Though, since it optimizes on 4 batches of data in a single step, it runs for one fourth the steps of MatFormer or OFA, and same number of steps as the baseline.

An important distinction between contribution of DynaBERT work and MatFormer is the search strategy. In MatFormer, we introduced the Mix'n'Match heuristic, which calculates an optimal submodel with a cost constraint without any overhead. In contrast, DynaBERT doesn't introduce any search strategy and discusses only using explicitly trained granularities as submodels.

### B.4  Comparison with Baselines

Here we do a more detailed discussion of comparison between MatFormer and the baselines used in Section 4.

1. **MatFormer vs trained from scratch Baseline**: Let Baseline-{S, M, L, XL} combined process 4X tokens, where each granularity is trained on X tokens independently. Since MatFormer maintains just one model, we train it for 4X tokens, in each step sampling either of MatLM-{S, M, L, XL}. As a result, the total compute and peak memory is still the same as Baseline since we effectively train MatLM-{S, M, L, XL} for X tokens each – matching Baseline training configuration. But since we maintain a single model, the shared parameters get trained for up to 4X tokens for free. For example, each time we train XL/L/M granularity, since S granularity is contained in them, it also receives gradients updates. Therefore, the MatLM-S effectively processes 4X tokens, MatLM-M processes 3X tokens, MatLM-L processes 2X tokens, and MatLM-XL processes X tokens. We can also see this translating to Validation loss (Figure 2a) – the smaller granularities outperform Baseline by a large margin. Whereas the larger granularity match in performance.

2. **MatFormer vs DynaBERT:** DynaBERT, similar to MatFormer, trains $g$ nested subnetworks, but differs critically in its training methodology. It performs joint optimization - simultaneously processing 4 batches of data - 1 batch through each {S, M, L, XL} granularity. The final loss being optimized is the average of each granularity's loss. Conversely, MatFormer samples and optimizes individual granularities in separate steps, resulting in four times more gradient updates for the same amount of data and compute. This difference in training methodology translates to a substantial performance gap: DynaBERT exhibits a significant 0.01 log perplexity gap compared to MatFormer on the 850M model. Even with a 15% increase in compute, DynaBERT still falls short of MatFormer's performance. Moreover, unlike MatFormer, which offers flexibility in submodel size selection through its Mix'n'Match approach, DynaBERT lacks a dedicated sampling strategy.

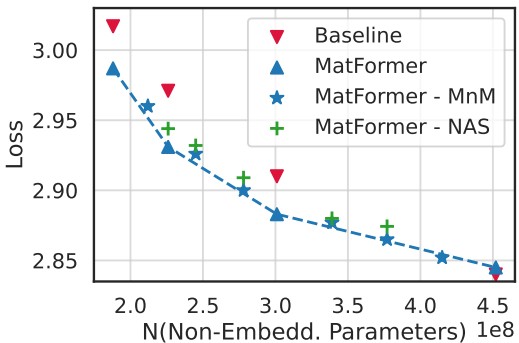

Figure 6: We search for optimal architectrure configuration using evolutionary search [48] and with Mix'n'Match. We find that Mix'n'Match yields architectures lying on pareto-optimal curve, and are at least as good as those found using the evolutionary search.

3. **MatFormer vs OFA:** OFA maintains one universal model similar to MatFormer, but randomly samples subnetworks such that each layer is one of {S, M, L, XL}. Similar to MatFormer and DynaBERT, it takes advantage of processing 4X data by a single model. But due to sampling, not enough small or large models (close to S and XL granularity model) are sampled, leading to inferior performance in these regions. That is, sampling models ranging from S model size to XL model size, most of the sampled models are close to M-L model sizes. This can also be seen in the Loss curve, where we see a bell like curve having much higher loss close to S and XL model sizes, and matching MatLM's performance in between. Another disadvantage of OFA is since random model are sampled, once can't employ simple Mix'n'Match technique like in matFormer. Rather a more complicated NAS strategy is needed to find optimal submodels. To train a NAS when searching for such large models is costly and erroneous, as we discuss in Appendix D.2.

## C   Training and Inference Costs

We currently make minimal changes and optimizations to the training scripts of vanilla Transformer architecture. In other words, we use the same implementation for both Baselime and MatFormer, except using different sized splices of FFN block for each forward pass. The wall-clock time for MatLM training is the same cost as training all the $4$ granulatities baseline counterparts. During serving, we observe the 850M model FFN latency to attention latency ratio $= 56 : 44$. We note that this FFN:MHA latency ratio depends highly on scale and sequence length. More specifically, for a given sequence length FFN latency dominates the overall latency at scale, while the attention heads' cost increases with sequence length. We refer the reader to Kim et al. [30] for a more extensive illustration of this. We emphasize that though we trained one MatFormer and compare its training time with Baselines combined, we can extract many more models than the 4 model granularities we explicitly trained.

### C.1   Speculative Decoding Attention Sharing

An additional benefit of MatLM is that the attention cache is shared between the draft and verifier model. When the XL model verifies S model's draft, it overwrites the attention cache with its richer latent representation compared to the one generated by the drafter model. Note that 1) this does not involve extra computation since MatLM has a single universal model including both draft and verifier model; 2) attention sharing isn't possible in the Baseline since they are not explicitly trained together. Hence, latent representation of one model is quite meaningless to the other model. Thus, attention sharing gives further improvement over vanilla speculative decoding as shown in Table 2.

Table 5: On 850M MatFormer model, while running NAS we observe it prefers balanced granularities across layers rather than skewed. On a few parameter constraints we list the MnM heuristic configuration, and configuration predicted with NAS.

| FFN Params budget | Mix'n'Match configuration & loss | NAS predicted configuration & loss |
|---|---|---|
| 226M | [M,M,M,M,M,M,M,M,M,M,M,M,M,M,M]; 2.931 | [S,M,M,M,M,M,M,S,M,M,M,M,M,M,L]; 2.944 |
| 245M | [M,M,M,M,M,M,M,M,M,M,M,M,L,L,L]; 2.926 | [M,M,M,M,M,L,M,M,M,L,M,L,M,M,M,L]; 2.932 |
| 278M | [M,M,M,M,M,L,L,L,L,L,L,L,L,L,L]; 2.9 | [M,L,L,L,L,L,L,L,L,L,M,L,M,M,M,L]; 2.91 |
| 377M | [L,L,L,L,XL,XL,XL,XL,XL,XL,XL,XL,XL,XL,XL,XL]; 2.865 | [L,L,XL,L,XL,L,XL,L,XL,XL,XL,L,XL,L,L,XL]; 2.874 |

# D   Search Techniques

## D.1   Mix'n'Match

For a given compute or parameter budget multiple submodels may meet the constraint. A common strategy could involve employing Neural Architecture Search (NAS) to identify the optimal architecture within this subset; however, this approach can be prohibitively expensive. Instead, we propose a simpler Mixn'Match heuristic, which identifies optimal submodels that lie on the Pareto-optimal accuracy-vs-parameters curve.

The Mix'n'Match heuristic recommends selecting layer granularities with minimal changes across layers, ensuring that the granularity of the $j^{th}$ layer is at least that of the $i^{th}$ layer for $j > i$. This means that a gradual increase in granularity with the "least slope" empirically yields the best performance among all tested mix-and-match configurations. Our heuristic is underpinned by the training methodology, where each sampled granularity configuration maintains consistent layer granularity across the model. Consequently, the model adapts best to configurations where layer granularities are either uniform or display minimal variation. For instance, a configuration of [M, M, M, M, L, L, L] across layers is more effective than [S, S, S, S, S, XL, XL] despite having similar number of parameters, as it maintains a more balanced distribution of granularity as opposed to a skewed one.

This intuition is also supported by results from an evolutionary search-based Neural Architecture Search (NAS) method [48], similar to that employed in [9]. Our NAS experiments, few of the runs we present in Table 5, indicated a preference for balanced configurations under various constraints, aligning with the findings from our Mix'n'Match heuristic.

We explored various balanced configurations, including:

1. Increasing-Decreasing: Granularity increases until the midpoint and then decreases, e.g., [M, M, L, L, L, M, M].

2. Decreasing-Increasing: The reverse of the previous heuristic, where granularity decreases then increases.

3. Increasing: A non-decreasing sequence of layer granularities, such as [M, M, M, M, L, L, L].

4. Decreasing: The opposite of the Increasing heuristic.

Among these configurations, the Increasing heuristic consistently outperformed the others. Therefore, we recommend adopting a strategy of selecting increasing granularities across layers with the minimum slope for effective model performance.

On the 850M MatLM, we compare architectures predicted by Mix'n'match and the NAS algorithm, in Figure 6. We find that our heuristic works at least as well as NAS, resulting in models that lie on pareto-optimal curve. Note that Mix'n'match requires no additional training of a module or overhead in calculating optimal submodel within a cost constraint.

## D.2   NAS

Neural Architecture Search [68, 48] is a natural method to search an optimal network among possible architectures within a constraint budget. In fact, OFA [9] uses an evolutionary search based technique [48] along with a architecture loss predictor (Appendix B.3) in its search routine. While on smaller scale models, it might be easy to deploy NAS method similar to what OFA does, on larger scale it becomes prohibitively costly and erroneous.

To get loss corresponding to an architecture, one needs to run on a sufficiently large held out set to get an average value. For example, in our runs we used a held out set of around 200M tokens. On large scale, it becomes costly to repeatedly run large models on this held out set to collect loss values. For example, to collect 16k (architecture, loss) pairs, we used 256 TPUv-4 chips for around 2 days.

Once we have the dataset, we split it into 60%-40% train-eval split. After training the architecture loss predictor, we tested out its performance on the held out set and noticed $\sim 5e - 5$ mean squared error.

As we move to large models regime, the performance gap between different sized models diminish. For example, the gap between 78M parameter Baseline-S and Baseline-XL, which are of size 71M and 78M, is 4.01 vs 3.83. Which is a gap of 0.18 loss for just 7M parameter difference. But in case of 850M model, Baseline-S and Baseline-XL has size 582M vs 850M , the corresponding loss is 3.017 vs 2.84. Thus, a difference of 0.177 for 268M parameter difference. This trend scales to larger models, and the gap reduces between various model sizes. So, training a architecture accuracy predictor becomes increasingly challenging and erroneous, since it has to learn differentiating architectures with very small differences in their loss values.

In our experiment for OFA, we take the best of the model configs predicted by NAS and the model config among the sampled points for a corresponding budget.

# E    Scaling Laws for Language Decoders

We provide results split by granularities for validation loss, average score on evaluation tasks, and consistency in Figures 9, 10, and 11 respectively. We observe that the gap in validation loss and evaluation accuracy between MatLMs and Baselines appears to be constant. For consistency, the gap appears to reduce with scale, but one would need to scale the models by many orders of magnitude beyond what's possible today for baselines to have comparable consistency with MatLMs.

## E.1    Scaling laws of MatFormers vs Transformers.

Scaling laws are essential tools to estimate quality as the cost of training or inference is increased. Scaling laws can help us consider various nuances of training and deployment such as overall training cost in FLOPS, training data and parameter efficiency, and inference mean FLOPS utilization vs latency for deployments.

The scaling relationship of MatFormers versus Transformers is both simple and complex. Simple, because MatFormers scaling curves for pretraining are similar to that of Transformers – thus Mat-Formers only require a similar amount of compute and the same hyperparameters that work for Transformers are effective for MatFormers. The complex scaling relationship comes from the fact that MatFormers allow the training of multiple models with a single training run which is a qualitative different from Transformers and difficult to factor into scaling equations. In terms of efficiency, if we compare the training FLOPs equivalent of all the extractable models from MatFormers, then MatFormer training alone has a clear advantage in any case where all parameters used to train standard Transformer models on the same dataset exceed $2.58P$, where $P$ is the number of parameters of the MatFormer and the largest Transformer model. This is so because MatFormer uses 2.58 times more FLOPs per token for a training run than a Transformers: $4\times$ more FLOPs for attention layers parameters and $\{1 + 1/2 + 1/4 + 1/8 = 1.875\}\times$ more FLOPs for MLP layers.

# F    Further Analysis on Language Decoders

## F.1    KL Divergence Between S, M, L and XL Models

Figure 7 showcases the smoother consistency calculation between two generative models measured with KL-divergence of the smaller model's outputs with the larger model outputs. Similar to the exact match style hard consistency metric used in the main paper, there is a significant gap between the consistency of MatLM's submodels with the MatLM-XL model and between that of the corresponding baseline models. This points to how sampling strategies based on the output probabilities do not change the behavioral consistency between two models and that it still follows the trend of generating the token with the highest probability. This smoother notion of consistency argues for the metric-space

Table 6: For 850M model, we experiment with modifying $\{p_S, p_M, p_L, p_{XL}\}$ to sample submodels from a non-uniform distribution during training and report the results across all granularities. We find that all strategies that upweight the loss for the largest granularity perform well, with modest degradation on the M and S granularties.

| Model | Probabilities | S | M | L | XL |
|-------|---------------|---|---|---|----|
| Baseline | N/A | 3.017 | 2.971 | 2.910 | 2.840 |
| MatFormer | 0.44/0.31/0.15/0.10 | **2.963** | **2.925** | 2.899 | 2.877 |
|  | 0.31/0.27/0.22/0.20 | 2.977 | 2.929 | 2.890 | 2.857 |
|  | 0.25/0.25/0.25/0.25 | 2.970 | 2.934 | 2.886 | 2.846 |
|  | 0.20/0.22/0.24/0.34 | 3.000 | 2.939 | 2.885 | 2.836 |
|  | 0.17/0.20/0.22/0.41 | 3.010 | 2.943 | **2.884** | **2.829** |

preservation given that the output classifier/embedding matrix is shared across all the submodels of MatLM.

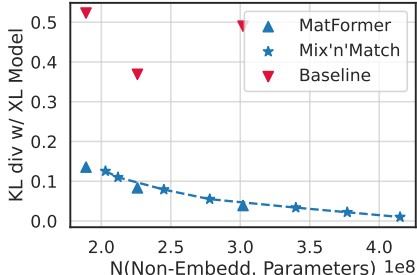

Figure 7: The smoother variant of consistency measures the KL divergence between the smaller models and the corresponding XL model. This metric, unlike the exact match accuracy variant, also accounts for different sampling strategies on the output distribution during deployment. In this figure, we plot KL divergence of S, M, L granularities with respect to XL for the 850M parameter model.

### F.2 Using MatFormers for Attention Sub-Block

We experiment with applying MRL to the attention sub-block of the Transformer block. More specifically, we jointly optimize 4 subnetworks with a varying number of attention heads $n_{attn}$, $3 * n_{attn}/4$, $n_{attn}/2$, $n_{attn}/4$ of size $d/n_{attn}$ each. In Figure 8, we plot the validation loss for these models (MatLM-Attn), their corresponding baselines and Mix'n'Matched subnetworks. We find that similar to our experiments on Mix'n'Matched MatLMs, Mix'n'Match helps obtain many MatLM-Attn models on the optimal loss-vs-compute curve.

### F.3 Tuning Sampling Probability

In Table 6, we experiment with tuning the sampling probabilities for individual granularities in order to investigate the trade-off between granularity size and performance. More specifically, we tune $\{p_1, p_2, p_3, p_4\}$, and find that all strategies that upweight the loss for the largest granularity perform well, with modest degradation on the M and S granularties.

## G Further Analysis on Vision Encoders

### G.1 Decoupling Effect of MatFormer on Pretraining and Finetuning

Table 7 investigates the effect of MatFormer on pretaining and finetuning phases of ViT-L/16 model. ViT-L/16 is typically pretrained on ImageNet-21K and then finetuned on ImageNet-1K for the final evaluation. Table 7 shows that having a MatFormer during pretraining generates a better model for

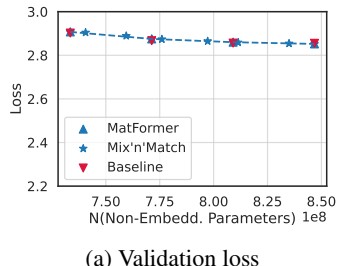

(a) Validation loss

Figure 8: Validation loss for the 850M MatLM-Attn & baseline models.

downstream finetuning compared to regular ViT pertaining. At the same time, finetuning a vanilla pretrained ViT with MatFormer results in flexibility being induced into the model. Despite being up to 2% less accurate than its counterparts at some granularities, a fine-tuned MatViT learned to reallocate the information to provide strong nested models. Considering that this is insignificant compared to pretaining costs, possible to take the largest pretrained ViT model and finetune with MatFormer to obtain a deployable MatViT variant.

Table 7: 2 × 2 grid of pairs to evaluate (top-1 accuracy (%)) the effects of MatFormer and standard training on the pretraining (PT) on ImageNet-21K and finetuning (FT) on ImageNet-1K using a L/16 architecture. Using a MatFormer during pretraining helps bring more accurate, and elastic encoders for downstream uses.

| PT↓ / FT→ | # Params (M) | ViT | MatViT |
|---|---|---|---|
| ViT | 306 | 85.26 | 85.57 |
| | 206 | 85.12 | 84.27 |
| | 156 | 85.02 | 82.79 |
| | 131 | 84.42 | 82.1 |
| MatViT | 306 | 85.58 | 85.61 |
| | 206 | – | 85.40 |
| | 156 | – | 85.02 |
| | 131 | – | 84.41 |

## G.2 Traditional Image Retrieval Evaluation

Table 8 showcases traditional image retrieval evaluation on ImageNet-1K where the query and the document encoders are the same for nearest neighbor retrieval. The 1-nearest neighbor (NN) based evaluation closely follows one-vs-all classification results shown in Figure 4. Both MatViT variants B/16 and L/16 have submodels that have as good or better retrieval performance compared to their independently trained counterparts. Concretely, MatViT-based retrieval can be up to 0.5% more accurate than the baselines while a 200M parameter MatViT submodel can be more accurate than the 300M parameter ViT baseline.

Table 8: Image retrieval 1-NN accuracy (%) when the query and document encoders are the same model. Similar to the image classification results, MatViT variants either match or outperform the corresponding standard ViT counterparts. Note that all the smaller models of a given model in MatViT are extracted for free while the baselines have to be explicitly trained for the constraints.

| Encoder | # Params (M) | ViT | MatViT |
|---------|--------------|-------|--------|
| B/16 | 85 | 77.46 | 77.38 |
|  | 57 | 76.58 | 76.41 |
|  | 43 | 74.90 | 74.49 |
|  | 36 | 71.44 | 71.72 |
| L/16 | 300 | 83.17 | 83.67 |
|  | 200 | 82.92 | 83.23 |
|  | 150 | 82.81 | 82.89 |
|  | 125 | 82.22 | 82.14 |

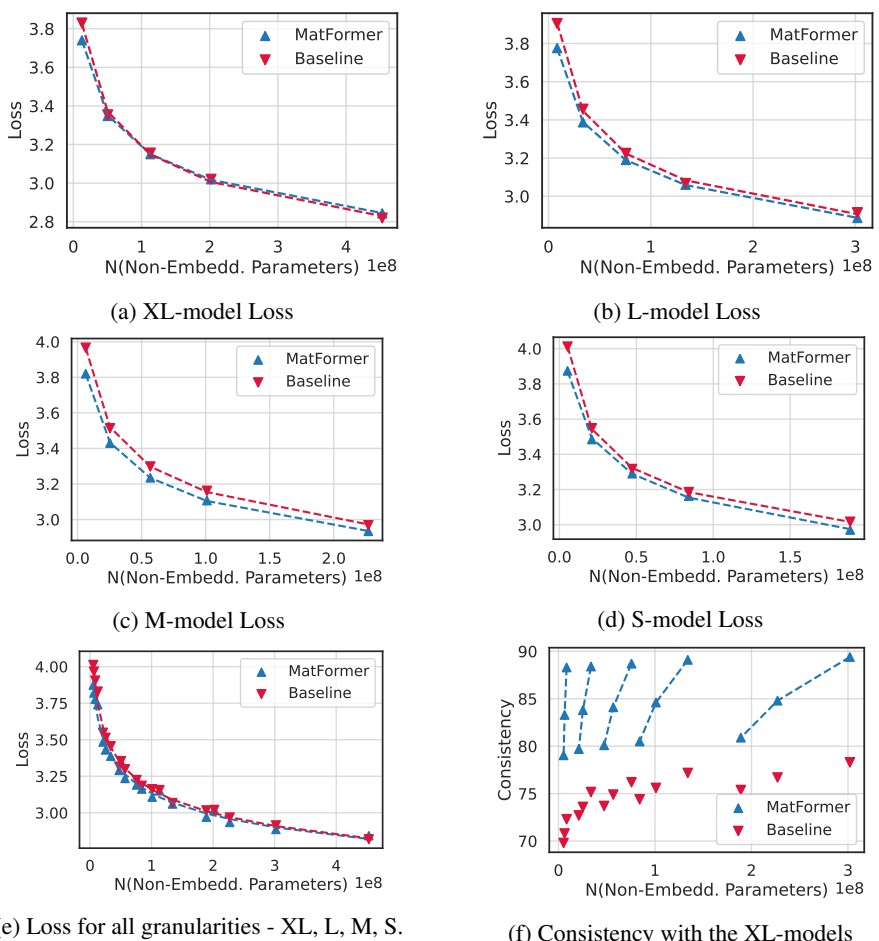

(a) XL-model Loss

(b) L-model Loss

(c) M-model Loss

(d) S-model Loss

(e) Loss for all granularities - XL, L, M, S.

(f) Consistency with the XL-models

Figure 9: We train various decoder-only MatLM models at a range of sizes from 78M to 850M parameters and observe the scaling trends for each model granularity on validation loss. We observe that the gap between MatLM and the baseline appears to be constant at each granularity. The consistency between the submodels of granularities and the XL models shows the effect of MatFormer joint training on natively ensuring similar behavior across submodels.

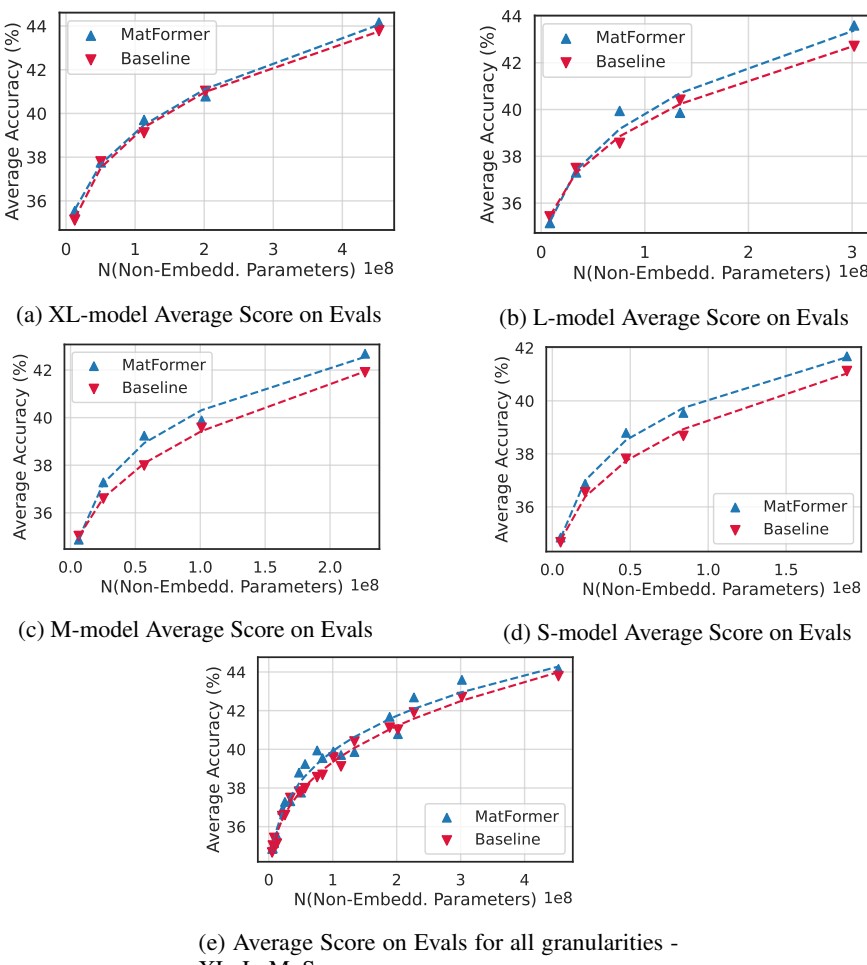

(a) XL-model Average Score on Evals

(b) L-model Average Score on Evals

(c) M-model Average Score on Evals

(d) S-model Average Score on Evals

(e) Average Score on Evals for all granularities -
XL, L, M, S

Figure 10: We train various decoder-only MatLM models at a range of sizes from 78M to 850M parameters and observe the scaling trends for each model granularity for the average score on 1-shot evaluation. We observe that the gap between MatLM and the baseline stays the same with scale, outperforming the baselines for on all granularities for the largest models.

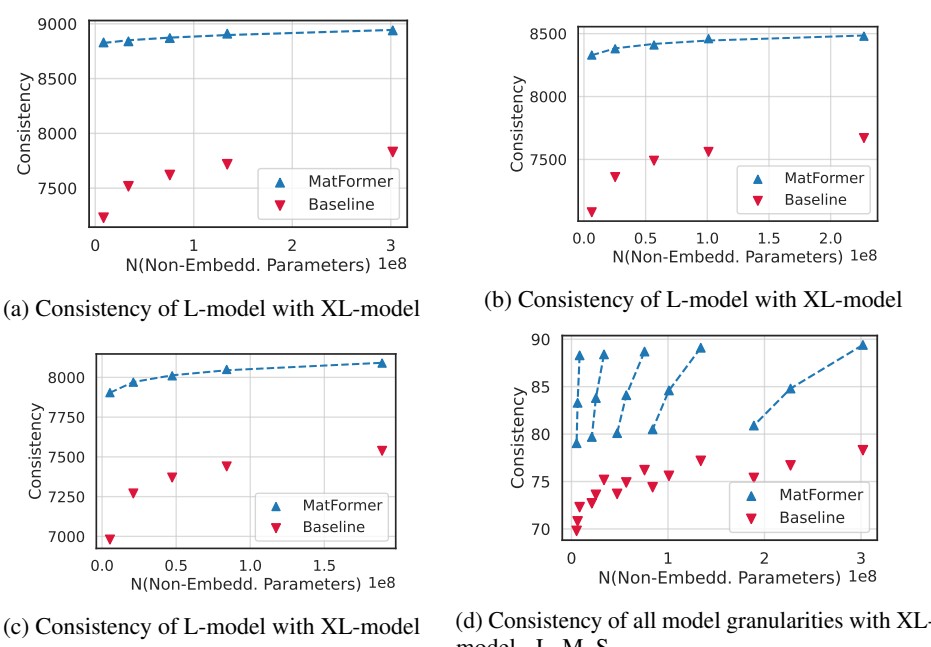

(a) Consistency of L-model with XL-model

(b) Consistency of L-model with XL-model

(c) Consistency of L-model with XL-model

(d) Consistency of all model granularities with XL-model - L, M, S

Figure 11: We train various decoder-only MatLM models at a range of sizes from 78M to 850M parameters and observe the scaling trends for each submodel S, M, L for the consistency with the XL model. We observe that the gap between MatLM and the baseline reduces with scale, but one would need to scale the baseline by many orders of magnitude to have consistency comparable to that of MatLMs.

Table 9: Downstream Eval numbers and development set log perplexity loss on 78M model size granularities.

| Downstream Task | Baseline-S | MatLM-S | Baseline-M | MatLM-M | Baseline-L | MatLM-L | Baseline-XL | MatLM-XL |
|---|---|---|---|---|---|---|---|---|
| TriviaQA (EM) | 0.14 | 0.13 | 0.19 | 0.14 | 0.14 | 0.11 | 0.19 | 0.10 |
| NaturalQuestions (EM) | 0.06 | 0.03 | 0.03 | 0.03 | 0.03 | 0.03 | 0.03 | 0.00 |
| WebQuestions (EM) | 0.10 | 0.10 | 0.15 | 0.10 | 0.20 | 0.15 | 0.30 | 0.15 |
| LAMBADA | 0.06 | 0.00 | 0.02 | 0.02 | 0.02 | 0.02 | 0.00 | 0.04 |
| HellaSwag | 25.42 | 26.59 | 26.00 | 26.28 | 25.95 | 25.95 | 25.95 | 26.32 |
| StoryCloze | 52.81 | 53.82 | 53.13 | 54.14 | 54.46 | 55.32 | 54.46 | 55.16 |
| WSC | 52.98 | 52.98 | 53.68 | 54.74 | 55.79 | 55.09 | 52.28 | 57.19 |
| WinoGrande | 48.46 | 47.91 | 51.54 | 50.36 | 50.99 | 51.38 | 48.86 | 50.20 |
| Winograd | 53.11 | 55.31 | 52.38 | 52.01 | 55.31 | 55.31 | 52.75 | 55.68 |
| RACE-H | 25.53 | 27.22 | 24.73 | 27.13 | 26.07 | 26.70 | 25.96 | 26.73 |
| RACE-M | 29.18 | 32.80 | 28.83 | 33.15 | 28.83 | 32.94 | 29.74 | 32.38 |
| PIQA | 55.77 | 56.47 | 54.62 | 56.31 | 54.52 | 56.42 | 56.86 | 56.09 |
| ARC-C | 21.50 | 22.01 | 21.08 | 21.50 | 21.59 | 21.76 | 22.35 | 21.76 |
| ARC-E | 34.55 | 34.55 | 34.30 | 34.68 | 34.89 | 34.97 | 34.55 | 34.55 |
| OpenBookQA | 25.40 | 28.00 | 27.60 | 29.00 | 28.20 | 30.60 | 29.80 | 30.60 |
| BoolQ | 48.72 | 52.14 | 51.87 | 52.08 | 51.28 | 52.14 | 52.11 | 52.20 |
| COPA | 62.00 | 58.00 | 62.00 | 59.00 | 63.00 | 58.00 | 60.00 | 59.00 |
| RTE | 53.79 | 53.43 | 52.35 | 53.07 | 51.26 | 52.35 | 51.99 | 52.35 |
| WiC | 49.53 | 47.49 | 49.06 | 47.18 | 47.34 | 47.34 | 47.65 | 47.18 |
| MultiRC | 51.28 | 53.38 | 51.51 | 53.96 | 47.67 | 53.57 | 49.38 | 52.74 |
| RECORD | 39.52 | 41.60 | 40.03 | 41.75 | 40.55 | 42.77 | 40.80 | 43.72 |
| CB | 41.07 | 41.07 | 44.64 | 41.07 | 44.64 | 41.07 | 42.86 | 42.86 |
| ANLI-RI | 30.90 | 32.60 | 32.30 | 32.30 | 32.50 | 32.40 | 32.50 | 32.70 |
| ANLI-R2 | 31.10 | 30.60 | 31.10 | 30.60 | 30.70 | 30.60 | 30.60 | 30.70 |
| ANLI-R3 | 31.75 | 31.17 | 30.58 | 30.58 | 30.33 | 30.67 | 30.00 | 31.00 |
| Average | 34.59 | 35.17 | 35.02 | 34.86 | 35.41 | 35.20 | 35.12 | 35.53 |
| Loss | 4.011 | 3.874 | 3.966 | 3.82 | 3.905 | 3.776 | 3.83 | 3.74 |

Table 10: Downstream Eval numbers and development set log perplexity loss on 180M model size granularities.

| Downstream Task | Baseline-S | MatLM-S | Baseline-M | MatLM-M | Baseline-L | MatLM-L | Baseline-XL | MatLM-XL |
|---|---|---|---|---|---|---|---|---|
| TriviaQA (EM) | 1.04 | 0.79 | 0.98 | 1.05 | 1.16 | 1.11 | 1.86 | 1.21 |
| NaturalQuestions (EM) | 0.08 | 0.17 | 0.14 | 0.30 | 0.30 | 0.33 | 0.28 | 0.39 |
| WebQuestions (EM) | 0.59 | 0.44 | 0.44 | 0.64 | 1.28 | 0.79 | 1.33 | 0.94 |
| LAMBADA | 0.16 | 1.13 | 0.43 | 1.28 | 1.51 | 1.20 | 0.49 | 1.28 |
| HellaSwag | 27.77 | 27.99 | 27.45 | 28.29 | 27.58 | 28.53 | 28.86 | 28.95 |
| StoryCloze | 56.33 | 57.03 | 57.03 | 57.40 | 57.30 | 58.31 | 58.63 | 58.90 |
| WSC | 55.44 | 56.14 | 56.49 | 57.89 | 58.25 | 58.25 | 57.54 | 57.89 |
| WinoGrande | 52.01 | 52.09 | 50.28 | 54.30 | 51.22 | 52.25 | 51.54 | 51.54 |
| Winograd | 54.21 | 56.78 | 56.78 | 56.78 | 61.54 | 57.14 | 60.44 | 61.17 |
| RACE-H | 27.93 | 28.82 | 27.50 | 28.47 | 28.70 | 29.33 | 28.73 | 28.99 |
| RACE-M | 33.29 | 34.05 | 34.19 | 33.98 | 34.54 | 34.89 | 33.29 | 35.58 |
| PIQA | 57.13 | 59.30 | 56.91 | 59.85 | 57.94 | 59.52 | 59.52 | 60.50 |
| ARC-C | 22.53 | 22.10 | 23.63 | 22.87 | 24.06 | 23.55 | 24.66 | 22.95 |
| ARC-E | 40.24 | 40.19 | 40.19 | 41.46 | 41.71 | 41.71 | 41.62 | 42.72 |
| OpenBookQA | 30.60 | 34.20 | 30.80 | 33.60 | 31.00 | 33.60 | 34.00 | 35.00 |
| BoolQ | 54.13 | 52.48 | 52.45 | 53.00 | 55.63 | 52.57 | 55.90 | 52.94 |
| COPA | 62.00 | 65.00 | 61.00 | 63.00 | 61.00 | 66.00 | 64.00 | 64.00 |
| RTE | 52.71 | 50.18 | 52.35 | 50.90 | 50.54 | 47.65 | 52.71 | 48.38 |
| WiC | 47.34 | 47.34 | 47.34 | 49.84 | 47.96 | 47.96 | 47.65 | 47.96 |
| MultiRC | 51.44 | 52.91 | 52.52 | 52.62 | 50.23 | 53.09 | 52.41 | 52.41 |
| RECORD | 48.58 | 50.56 | 48.99 | 52.11 | 50.56 | 53.21 | 52.82 | 54.09 |
| CB | 42.86 | 42.86 | 42.86 | 42.86 | 39.29 | 44.64 | 42.86 | 42.86 |
| ANLI-RI | 31.80 | 32.20 | 31.80 | 31.50 | 32.40 | 32.50 | 32.20 | 32.60 |
| ANLI-R2 | 30.50 | 29.40 | 31.10 | 29.70 | 32.00 | 29.30 | 30.50 | 30.30 |
| ANLI-R3 | 30.08 | 31.33 | 30.50 | 30.67 | 33.50 | 30.75 | 30.67 | 30.42 |
| Average | 36.43 | 37.02 | 36.56 | 37.37 | 37.24 | 37.57 | 37.78 | 37.75 |
| Loss | 3.548 | 3.484 | 3.513 | 3.43 | 3.456 | 3.387 | 3.354 | 3.348 |

Table 11: Downstream Eval numbers and development set log perplexity loss on 310M model size granularities.

| Downstream Task | Baseline-S | MatLM-S | Baseline-M | MatLM-M | Baseline-L | MatLM-L | Baseline-XL | MatLM-XL |
|---|---|---|---|---|---|---|---|---|
| TriviaQA (EM) | 2.09 | 3.77 | 2.20 | 3.87 | 2.84 | 4.92 | 5.18 | 5.40 |
| NaturalQuestions (EM) | 0.11 | 0.69 | 0.28 | 0.72 | 0.58 | 0.72 | 0.91 | 1.00 |
| WebQuestions (EM) | 2.12 | 2.41 | 1.08 | 2.41 | 1.67 | 2.71 | 2.41 | 3.35 |
| LAMBADA | 0.29 | 1.28 | 0.66 | 2.10 | 1.90 | 2.66 | 2.76 | 3.10 |
| HellaSwag | 29.89 | 30.49 | 30.05 | 31.28 | 31.18 | 31.87 | 32.52 | 32.65 |
| StoryCloze | 59.17 | 60.24 | 59.54 | 60.88 | 60.24 | 61.79 | 61.68 | 62.48 |
| WSC | 61.05 | 59.30 | 59.30 | 60.00 | 61.75 | 63.16 | 58.95 | 62.46 |
| WinoGrande | 51.46 | 52.25 | 49.57 | 50.75 | 52.41 | 52.72 | 50.91 | 52.33 |
| Winograd | 55.68 | 62.27 | 57.88 | 64.10 | 63.00 | 67.77 | 61.17 | 67.40 |
| RACE-H | 29.45 | 29.50 | 28.90 | 29.42 | 29.22 | 29.87 | 29.67 | 29.42 |
| RACE-M | 35.31 | 37.19 | 36.14 | 37.26 | 36.42 | 37.53 | 37.60 | 38.23 |
| PIQA | 58.98 | 60.83 | 59.58 | 61.92 | 59.79 | 63.00 | 62.19 | 62.62 |
| ARC-C | 23.38 | 25.17 | 23.21 | 24.06 | 23.81 | 24.49 | 25.00 | 23.98 |
| ARC-E | 42.30 | 43.86 | 44.11 | 45.50 | 44.53 | 47.05 | 46.80 | 48.36 |
| OpenBookQA | 32.80 | 35.60 | 34.60 | 35.80 | 35.20 | 37.80 | 36.80 | 37.00 |
| BoolQ | 53.43 | 54.56 | 55.32 | 53.79 | 52.87 | 51.87 | 54.22 | 51.31 |
| COPA | 61.00 | 62.00 | 61.00 | 62.00 | 64.00 | 67.00 | 60.00 | 65.00 |
| RTE | 52.71 | 49.10 | 53.43 | 49.46 | 51.62 | 48.74 | 54.15 | 51.26 |
| WiC | 47.18 | 48.75 | 47.65 | 47.65 | 47.65 | 47.34 | 47.34 | 47.34 |
| MultiRC | 51.67 | 51.57 | 52.70 | 51.16 | 53.84 | 52.58 | 53.28 | 52.76 |
| RECORD | 54.34 | 55.93 | 55.18 | 56.80 | 56.75 | 58.21 | 58.39 | 58.97 |
| CB | 42.86 | 44.64 | 42.86 | 51.79 | 42.86 | 50.00 | 50.00 | 44.64 |
| ANLI-RI | 32.00 | 33.20 | 32.00 | 33.40 | 32.50 | 33.30 | 32.20 | 32.40 |
| ANLI-R2 | 32.60 | 30.60 | 30.90 | 30.10 | 30.60 | 30.70 | 29.80 | 30.60 |
| ANLI-R3 | 32.08 | 31.58 | 30.75 | 31.83 | 32.17 | 32.00 | 31.50 | 31.17 |
| Average | 37.75 | 38.67 | 37.95 | 39.12 | 38.77 | 40.00 | 39.41 | 39.81 |
| Loss | 3.316 | 3.29 | 3.299 | 3.235 | 3.225 | 3.19 | 3.16 | 3.15 |


Table 12: Downstream Eval numbers and development set log perplexity loss on 463M model size granularities.

| Downstream Task | Baseline-S | MatLM-S | Baseline-M | MatLM-M | Baseline-L | MatLM-L | Baseline-XL | MatLM-XL |
|---|---|---|---|---|---|---|---|---|
| TriviaQA (EM) | 4.63 | 5.67 | 4.87 | 6.78 | 6.11 | 7.51 | 8.09 | 7.76 |
| NaturalQuestions (EM) | 0.61 | 1.19 | 0.80 | 1.33 | 0.94 | 1.41 | 1.66 | 1.33 |
| WebQuestions (EM) | 2.31 | 2.61 | 2.26 | 2.56 | 2.85 | 2.71 | 2.85 | 3.20 |
| LAMBADA | 2.10 | 1.82 | 2.60 | 1.98 | 3.94 | 2.60 | 3.49 | 3.71 |
| HellaSwag | 32.12 | 32.69 | 32.83 | 34.05 | 33.80 | 34.88 | 36.21 | 36.36 |
| StoryCloze | 61.25 | 61.46 | 61.36 | 62.21 | 63.66 | 63.23 | 64.24 | 64.35 |
| WSC | 57.54 | 62.81 | 61.40 | 62.11 | 66.32 | 62.46 | 61.05 | 62.81 |
| WinoGrande | 52.33 | 51.46 | 49.09 | 51.78 | 52.64 | 50.20 | 53.12 | 51.85 |
| Winograd | 60.07 | 61.17 | 60.07 | 63.37 | 67.40 | 64.47 | 68.50 | 64.84 |
| RACE-H | 29.85 | 29.65 | 29.47 | 30.10 | 30.56 | 30.33 | 30.70 | 30.93 |
| RACE-M | 37.53 | 38.23 | 37.33 | 39.28 | 40.39 | 39.62 | 40.95 | 40.67 |
| PIQA | 61.26 | 61.43 | 61.48 | 62.40 | 60.99 | 62.79 | 63.17 | 63.44 |
| ARC-C | 23.04 | 24.23 | 24.06 | 24.32 | 24.49 | 25.43 | 23.72 | 24.91 |
| ARC-E | 45.83 | 46.42 | 46.30 | 47.35 | 47.73 | 49.54 | 51.73 | 50.72 |
| OpenBookQA | 37.20 | 37.40 | 37.00 | 38.60 | 36.40 | 38.80 | 41.00 | 39.20 |
| BoolQ | 52.39 | 51.77 | 56.12 | 51.96 | 50.28 | 53.52 | 54.98 | 55.17 |
| COPA | 67.00 | 65.00 | 73.00 | 64.00 | 71.00 | 63.00 | 67.00 | 70.00 |
| RTE | 52.35 | 55.23 | 53.43 | 50.54 | 52.35 | 49.46 | 52.35 | 52.35 |
| WiC | 47.34 | 47.34 | 47.34 | 47.34 | 47.34 | 47.34 | 47.34 | 47.34 |
| MultiRC | 54.93 | 53.01 | 50.89 | 53.11 | 52.62 | 52.10 | 52.33 | 52.89 |
| RECORD | 57.58 | 59.93 | 59.31 | 61.06 | 60.87 | 62.16 | 63.42 | 63.51 |
| CB | 42.86 | 41.07 | 44.64 | 46.43 | 44.64 | 37.50 | 42.86 | 35.71 |
| ANLI-RI | 32.60 | 32.90 | 31.70 | 32.30 | 31.40 | 32.40 | 32.50 | 32.30 |
| ANLI-R2 | 30.70 | 33.70 | 28.40 | 30.50 | 30.40 | 30.90 | 31.20 | 31.90 |
| ANLI-R3 | 30.83 | 33.17 | 30.08 | 32.50 | 30.83 | 31.58 | 30.92 | 31.25 |
| Average | 39.05 | 39.64 | 39.43 | 39.91 | 40.40 | 39.83 | 41.01 | 40.83 |
| Loss | 3.205 | 3.162 | 3.16 | 3.107 | 3.096 | 3.06 | 3.023 | 3.02 |

Table 13: Downstream Eval numbers and development set log perplexity loss on 850M model size granularities.

| Downstream Task | Baseline-S | MatLM-S | Baseline-M | MatLM-M | Baseline-L | MatLM-L | Baseline-XL | MatLM-XL |
|---|---|---|---|---|---|---|---|---|
| TriviaQA (EM) | 6.62 | 10.50 | 9.78 | 11.61 | 11.72 | 13.15 | 13.31 | 15.15 |
| NaturalQuestions (EM) | 0.89 | 1.97 | 1.58 | 1.97 | 2.38 | 2.33 | 2.66 | 2.52 |
| WebQuestions (EM) | 3.35 | 3.69 | 4.18 | 2.95 | 4.43 | 3.94 | 4.08 | 4.82 |
| LAMBADA | 8.25 | 6.89 | 10.83 | 7.51 | 10.44 | 9.55 | 14.03 | 9.96 |
| HellaSwag | 36.64 | 37.42 | 37.70 | 39.21 | 39.64 | 41.48 | 43.40 | 43.43 |
| StoryCloze | 65.26 | 65.10 | 66.17 | 67.08 | 67.13 | 68.47 | 71.25 | 70.12 |
| WSC | 65.96 | 65.96 | 64.21 | 67.02 | 69.12 | 69.82 | 70.53 | 67.72 |
| WinoGrande | 51.54 | 52.57 | 52.57 | 54.46 | 52.96 | 54.54 | 54.14 | 54.85 |
| Winograd | 69.23 | 64.84 | 71.43 | 67.77 | 70.33 | 71.79 | 72.16 | 73.26 |
| RACE-H | 30.76 | 31.82 | 31.88 | 32.96 | 31.88 | 33.45 | 33.79 | 33.42 |
| RACE-M | 40.95 | 41.85 | 41.16 | 42.76 | 42.55 | 44.43 | 44.64 | 44.57 |
| PIQA | 63.98 | 64.04 | 64.91 | 65.02 | 65.23 | 66.81 | 67.25 | 67.19 |
| ARC-C | 24.15 | 26.02 | 24.91 | 27.65 | 26.54 | 28.84 | 27.13 | 30.38 |
| ARC-E | 51.01 | 52.69 | 52.95 | 55.13 | 54.92 | 56.27 | 57.11 | 59.01 |
| OpenBookQA | 40.40 | 41.60 | 41.20 | 42.20 | 40.80 | 42.20 | 43.00 | 43.40 |
| BoolQ | 50.31 | 52.48 | 47.80 | 54.80 | 50.15 | 54.59 | 55.60 | 54.46 |
| COPA | 73.00 | 74.00 | 73.00 | 73.00 | 73.00 | 75.00 | 73.00 | 78.00 |
| RTE | 51.99 | 54.87 | 52.35 | 54.15 | 51.99 | 54.15 | 53.07 | 53.43 |
| WiC | 47.18 | 47.34 | 47.18 | 47.34 | 47.18 | 47.18 | 47.34 | 47.18 |
| MultiRC | 51.79 | 55.01 | 53.09 | 54.81 | 53.36 | 55.61 | 56.08 | 56.19 |
| RECORD | 64.27 | 65.16 | 65.36 | 65.99 | 66.53 | 67.74 | 69.56 | 68.72 |
| CB | 37.50 | 41.07 | 42.86 | 44.64 | 42.86 | 48.21 | 46.43 | 50.00 |
| ANLI-RI | 31.80 | 32.70 | 32.10 | 31.80 | 32.20 | 31.20 | 32.60 | 31.30 |
| ANLI-R2 | 31.50 | 30.20 | 30.90 | 30.00 | 30.60 | 30.40 | 30.40 | 30.90 |
| ANLI-R3 | 30.25 | 30.92 | 30.17 | 31.33 | 30.00 | 32.08 | 30.58 | 31.92 |
| Average | 42.58 | 43.34 | 43.35 | 44.23 | 44.01 | 45.42 | 45.83 | 46.11 |
| ANLI-R3 | 30.83 | 33.17 | 30.08 | 32.50 | 30.83 | 31.58 | 30.92 | 31.25 |
| Average | 41.14 | 42.03 | 42.01 | 42.92 | 42.72 | 44.13 | 44.53 | 44.87 |
| Loss | 3.017 | 2.987 | 2.971 | 2.934 | 2.91 | 2.886 | 2.84 | 2.843 |

Table 14: Downstream Eval numbers and development set log perplexity loss on 850M model size granularities for DynaBERT and OFA

| Downstream Task | DynaBERT-S | OFA-S | DynaBERT-M | OFA-M | DynaBERT-L | OFA-L | DynaBERT-XL | OFA-XL |
|---|---|---|---|---|---|---|---|---|
| TriviaQA (EM) | 8.52 | 9.96 | 9.57 | 10.60 | 13.14 | 11.35 | 14.55 | 13.34 |
| NaturalQuestions (EM) | 1.39 | 1.52 | 1.83 | 2.24 | 2.55 | 2.22 | 2.96 | 2.60 |
| WebQuestions (EM) | 3.30 | 4.28 | 3.74 | 4.63 | 4.68 | 4.33 | 5.46 | 4.72 |
| LAMBADA | 8.85 | 4.72 | 11.84 | 5.74 | 14.88 | 5.98 | 14.65 | 8.31 |
| HellaSwag | 37.86 | 37.22 | 39.26 | 39.33 | 41.15 | 41.26 | 43.39 | 43.04 |
| StoryCloze | 65.42 | 64.40 | 66.54 | 66.49 | 68.36 | 68.57 | 69.05 | 69.75 |
| WSC | 64.91 | 68.07 | 68.77 | 65.96 | 71.58 | 68.77 | 72.63 | 68.42 |
| WinoGrande | 55.72 | 52.09 | 55.72 | 56.04 | 57.30 | 55.33 | 59.35 | 55.33 |
| Winograd | 69.23 | 68.86 | 69.60 | 68.50 | 72.53 | 70.33 | 75.46 | 72.53 |
| RACE-H | 31.16 | 32.70 | 32.30 | 32.96 | 32.62 | 32.99 | 33.19 | 33.96 |
| RACE-M | 41.43 | 42.41 | 43.04 | 43.04 | 45.54 | 43.73 | 46.52 | 43.73 |
| PIQA | 63.71 | 63.49 | 65.67 | 65.23 | 65.89 | 66.49 | 67.14 | 66.76 |
| ARC-C | 24.66 | 24.74 | 25.94 | 26.62 | 26.62 | 27.22 | 28.50 | 29.27 |
| ARC-E | 52.44 | 52.78 | 54.46 | 54.46 | 57.41 | 56.78 | 58.54 | 58.96 |
| OpenBookQA | 39.60 | 39.60 | 39.40 | 42.00 | 41.40 | 42.40 | 42.00 | 42.40 |
| BoolQ | 54.56 | 52.11 | 52.81 | 52.17 | 50.55 | 54.13 | 51.28 | 54.92 |
| COPA | 70.00 | 68.00 | 72.00 | 72.00 | 75.00 | 71.00 | 73.00 | 74.00 |
| RTE | 53.07 | 52.71 | 54.15 | 52.71 | 49.10 | 52.71 | 46.21 | 52.71 |
| WiC | 47.34 | 47.34 | 47.34 | 47.34 | 47.02 | 47.34 | 47.34 | 47.34 |
| MultiRC | 51.73 | 54.41 | 53.01 | 54.93 | 55.42 | 55.18 | 55.18 | 55.42 |
| RECORD | 65.05 | 64.60 | 66.65 | 66.61 | 68.33 | 67.96 | 68.91 | 68.68 |
| CB | 41.07 | 42.86 | 41.07 | 42.86 | 41.07 | 46.43 | 41.07 | 44.64 |
| ANLI-RI | 32.60 | 32.50 | 31.70 | 32.50 | 32.60 | 32.30 | 32.10 | 32.10 |
| ANLI-R2 | 30.60 | 30.60 | 30.60 | 30.50 | 31.00 | 30.80 | 31.70 | 30.70 |
| ANLI-R3 | 30.42 | 30.67 | 31.08 | 30.67 | 32.00 | 30.58 | 31.58 | 30.75 |
| Average | 41.78 | 41.71 | 42.72 | 42.64 | 43.9 | 43.44 | 44.47 | 44.17 |
| Loss | 2.993 | 3.01 | 2.942 | 2.935 | 2.895 | 2.89 | 2.854 | 2.863 |

Question: Does the paper discuss the limitations of the work performed by the authors?

Answer: [Yes]

Justification: We discuss this throughout the draft.

3. **Theory Assumptions and Proofs**

Question: For each theoretical result, does the paper provide the full set of assumptions and a complete (and correct) proof?

Answer: [NA] .

Justification: No theoretical results.

4. **Experimental Result Reproducibility**

Question: Does the paper fully disclose all the information needed to reproduce the main experimental results of the paper to the extent that it affects the main claims and/or conclusions of the paper (regardless of whether the code and data are provided or not)?

Answer: [Yes]

Justification: Implementation details are provided in Appendix B. Moreover, code to reproduce Section 4.2 will be been open-sourced at link.

5. **Open access to data and code**

Question: Does the paper provide open access to the data and code, with sufficient instructions to faithfully reproduce the main experimental results, as described in supplemental material?

Answer: [No]

Justification: Implementation details are provided in Appendix B. While code to reproduce Section 4.2 has been open-sourced, the language model has been trained on proprietary data.

6. **Experimental Setting/Details**

Question: Does the paper specify all the training and test details (e.g., data splits, hyper-parameters, how they were chosen, type of optimizer, etc.) necessary to understand the results?

Answer: [Yes]

Justification: Implementation details are provided in Appendix B, with citation of the models upon which these models are built.

7. **Experiment Statistical Significance**

   Question: Does the paper report error bars suitably and correctly defined or other appropriate information about the statistical significance of the experiments?

   Answer: [No]

   Justification: This has not been done due to the computational expense of training language models from scratch.

8. **Experiments Compute Resources**

   Question: For each experiment, does the paper provide sufficient information on the computer resources (type of compute workers, memory, time of execution) needed to reproduce the experiments?

   Answer: [Yes]

   Justification: This information is provided in Appendix B.

9. **Code Of Ethics**

   Question: Does the research conducted in the paper conform, in every respect, with the NeurIPS Code of Ethics https://neurips.cc/public/EthicsGuidelines?

   Answer: [Yes]

   Justification: Broader impacts have been discussed in Appendix A.

10. **Broader Impacts**

    Question: Does the paper discuss both potential positive societal impacts and negative societal impacts of the work performed?

    Answer: [Yes]

    Justification: Broader impacts have been discussed in Appendix A.

11. **Safeguards**

    Question: Does the paper describe safeguards that have been put in place for responsible release of data or models that have a high risk for misuse (e.g., pretrained language models, image generators, or scraped datasets)?

    Answer: [NA]

    Justification: No datasets or models are released with this paper.

12. **Licenses for existing assets**

    Question: Are the creators or original owners of assets (e.g., code, data, models), used in the paper, properly credited and are the license and terms of use explicitly mentioned and properly respected?

    Answer: [Yes]

    Justification: The vision models and data used in this have been public for a long time. The language models inherit the appropriate licenses of the Lamda [58] paper while we train on proprietary data.

13. **New Assets**

    Question: Are new assets introduced in the paper well documented and is the documentation provided alongside the assets?

    Answer: [Yes]

    Justification: Code to reproduce experiments will be released for camera ready.

    Guidelines:

    - The answer NA means that the paper does not release new assets.
    - Researchers should communicate the details of the dataset/code/model as part of their submissions via structured templates. This includes details about training, license, limitations, etc.

- The paper should discuss whether and how consent was obtained from people whose asset is used.
- At submission time, remember to anonymize your assets (if applicable). You can either create an anonymized URL or include an anonymized zip file.

14. **Crowdsourcing and Research with Human Subjects**

    Question: For crowdsourcing experiments and research with human subjects, does the paper include the full text of instructions given to participants and screenshots, if applicable, as well as details about compensation (if any)?

    Answer: [NA]

    Justification: The paper does not involve crowdsourcing nor research with human subjects.

    Guidelines:

    - The answer NA means that the paper does not involve crowdsourcing nor research with human subjects.
    - Including this information in the supplemental material is fine, but if the main contribution of the paper involves human subjects, then as much detail as possible should be included in the main paper.
    - According to the NeurIPS Code of Ethics, workers involved in data collection, curation, or other labor should be paid at least the minimum wage in the country of the data collector.

15. **Institutional Review Board (IRB) Approvals or Equivalent for Research with Human Subjects**

    Question: Does the paper describe potential risks incurred by study participants, whether such risks were disclosed to the subjects, and whether Institutional Review Board (IRB) approvals (or an equivalent approval/review based on the requirements of your country or institution) were obtained?

    Answer: [NA]

    Justification: The paper does not involve crowdsourcing nor research with human subjects.

    Guidelines:

    - The answer NA means that the paper does not involve crowdsourcing nor research with human subjects.
    - Depending on the country in which research is conducted, IRB approval (or equivalent) may be required for any human subjects research. If you obtained IRB approval, you should clearly state this in the paper.
    - We recognize that the procedures for this may vary significantly between institutions and locations, and we expect authors to adhere to the NeurIPS Code of Ethics and the guidelines for their institution.
    - For initial submissions, do not include any information that would break anonymity (if applicable), such as the institution conducting the review.

