# OpenReview forum: "MatFormer: Nested Transformer for Elastic Inference"
_NeurIPS.cc/2024/Conference — NeurIPS 2024 poster_

### Official Review · Reviewer_6C9L · 2024-06-12

**Soundness:** 3
**Presentation:** 3
**Contribution:** 3
**Rating:** 5
**Confidence:** 4

**Summary:**

The paper proposes Matformer, a technique to achieve elastic inference where one model is trained, encapsulating sub-models that can be extracted on demand. The main idea to achieve this is to apply a Matryoshka "self-stacking" of hidden states in the FNN blocks of transformers, which are randomly sampled at train time to for all sub-models to be trained concurrently. Experimental results on Million-parameter transformers show that this idea works very well in producing low complexity models at inference time.

**Strengths:**

The idea proposed is very original: train one model and obtain many sub-models at inference time.

Experimentally, the approach is shown to work well with LLMs and ViTs, and shows promising results in terms of scaling with vanilla transformers. In addition to compression, it is shown that an actual speed-up in execution can be achieved in tandem with speculative decoding.

Finally, the paper is very well written; thank you to the authors for making everything so clear that I was able to understand everything in my review.

**Weaknesses:**

I have identified a few weaknesses in the paper, most of which are included in the "questions" section below. Please address those:
- I have doubts on the application of the Matryoshka approach to the attention block (see questions)
- In terms of training time, it is unclear that the proposed approach would be better than training small models, when taking into account FLOPs per iteration and speed of convergence (larger models need more tokens).
- Experiments are limited to models with less than a billion parameters, while the description uses Llama 34B, 40B, and 70B as motivating examples.
-Minor: There are no theoretical results.

**Questions:**

I have several questions to the authors:

Issues in the motivation of the work:
- The authors identify compression/pruning as one option to fit a bigger model to a compute budget, but argue that such scheme requires additional training. This is not very fair, since Matformer itself trains from scratch, one could also do quantization-aware, pruning-aware, compression-aware training from scratch. While this approach would still necessitate one trianing session for one model size, I disagree with the author's assertion that compression "requires additional training".
- At line 40, the authors claim that Matformer is applied to both attention and FFN blocks. However, Figure 1 shows that this is only applied to the hidden state of the FFN, and also later sections in the work (Section 3 where the scheme is presented, and Section 4 where results are reported) only use Matformwer on the FFN blocks. It seems like this in an overclaim in the introduction which should be removed. Unless Matformer can be applied to the Attention. If so, how do we define Matformer for Attention? In the FFN, it is very clear how the hidden state is arbitrarily truncated to smaller chunks in a Matryoshka fasion. What is the corresponding operation in Attention, is it applied to the attention heads? How would that impact LLM serving optimization such as KV cache optimization, FlashAttantion and etc...? And how would it differ from grouped query attention? A very vague hint to the potential of doing that is given at lines 147. Please discuss further, or just drop the issue of attention and explicitly state that the method is applied to FNN only. Right now, it's a bit confusing what is or can be done for the attention block.


On the related work section: Overall this is a good survey of similar works. I would suggest comparing to Flextron [1]. This work essentially does the same thing as Matformer as far as I can tell (authors, please let me know what the differences are). It is contemporary since it came out around the same time as the Neurips deadline, so this does not in any way impact my recommendation score for this paper. However, I think it is still good to compare to Flextron, who does elastic inference on Billion-parameter models.
[1] Flextron: Many-in-One Flexible Large Language Model, Cai et al., ICML2024


Questions on the training scheme proposed:
- Does the random sampling strategy affect convergence time? Since at each iteration, we are sampling one of exponentially many sub-networks, does it take more time to train (in terms of iterations or "epochs").
- How does the computational cost of training (measured in FLOPs) compares to training small models from scratch. I'd assume that training a small model from scratch would converge faster since parameter volume is known to be correlated with training token availability. And also, a smaller model would have a smaller computational cost per iteration. So, all in all, is it really easier to use Matformer, rather than training small models. I would appreciate a quantitative answer to this question, if possible.
P.S., the example of Llama3 34B, 40B, 80B is specifically talking to what I am referring to (even though it doesn't take into account the algorithmic convergence). We need to train for a number of FLOPs that is needed by Llama3-80B in order to produce a 40B model.


Comments on the experiments:
- In the earlier sections, a lot is being made about Llama models, yet all reported results are on Million-parameter models. I understand that Meta did not provide a training recipe for Llamas (nor a training dataset) - but there are open source implementations of Billion-parameter models, such as Megatron-LM. It would be good to see how Matformer scales beyond 850M parameters. E.g., Flextron does produce elastic inferencing up to 8B parameters. Currently, while all results are good, I am worried that using such small models may be deriving conclusions from toy examples.

**Limitations:**

Yes, there are no worries on that, and the authors did attach the Neurips checklist to the back of their paper.

---

> ### Author Rebuttal · Authors · 2024-08-06
>
> We thank the reviewer for their thoughtful comments and support. We clarify the main concerns below:
>
> 1."...unclear that the proposed approach would be better than training small..."
>
> We clarify that MatFormer does not use more data or compute compared to the baselines trained separately and can be scaled like the baselines. Rather than training different sized baseline models from scratch and separately, our approach trains them together and enjoys the benefits of training common parameters jointly.
>
> Consider 4 baselines trained using FLOPs-S/M/L/XL for Baseline-S/M/L/XL, resulting in FLOPs_TOTAL=FLOPs_S+...+FLOPs_XL. These separate baselines, collectively use memory Memory-S+Memory-M+Memory-L+Memory-XL. MatFormer, in comparison, has memory=max(Memory-S,...,Memory-XL)=Memory-XL. The FLOPs used by MatFormer does not exceed FLOPs-TOTAL.
>
> Moreover, Baselines offer only the fixed few models they were explicitly trained for, whereas MatFormer provides thousands of accurate models that were never explicitly trained for.
>
> 3. "...I disagree...that compression "requires additional training"."
>
> We'd like to clarify what we meant by “additional training”. MatFormer, once trained (using the same FLOPs as the 4 baseline models), provides 1000s of models that can be extracted without any post-hoc training. In contrast, with compression, after training a large model, additional post-hoc processes are required to obtain smaller models for each new latency budget. To generate 1000s of models, each of these processes would need to be repeated individually. We will add this nuance in the final draft.
>
> 4. "At line 40, the authors claim that Matformer is applied to both attention and FFN..."
>
> We have preliminary results on applying MatFormer to the attention block in Figure 8 (Appendix F2). Similar to the FFN case, we can get many submodels for free using Mix’n’Match. As you correctly referenced, we apply MatFormer structure to the number of attention heads n. Specifically, we use the “first” n/8 heads in MatFormer-S, first n/4 of heads (superset of n/8 heads) in MatFormer-M, first n/2 heads in MatFormer-L and all n heads in MatFormer-XL. This results in the attention KV cache being reduced proportionally—⅛ / ¼ / ½ for MatFormer-S/M/L. Optimization is the same as optimizing a smaller model having n/8, n/4, or n/2 heads respectively. Flash Attention works on individual heads independently, so it will act on whatever heads are being considered for a given model. For grouped query attention, MatFormer could be applied group-wise, prioritizing the first few groups and so on.
>
> 5. "...would suggest comparing to Flextron [1]..."
>
> We thank the reviewer for their feedback, and will include these details in the final draft.
>
> 6. "Does the random sampling .. convergence time .. sampling one of exponentially many sub-networks..."
>
> We would like to clarify that we sample only one of g models per step, not one of the exponentially many possible sub-networks. Convergence is not affected by this. We have also experimented with alternative sampling methods, such as rotating sequentially through Model-S, M, L, and XL. These methods yielded similar performance to random sampling. The total training time for MatFormer is always less than or equal to the time required to train the individual baseline models separately.
>
> 6. "How does the computational cost of training (measured in FLOPs) compares to training..."
>
> Consider the Llama3 model family as an example —34B, 40B, 80B models. Assume each model is trained for X, Y, and Z steps with the same batch size.
>
> In contrast, MatFormer trains a single 80B model, thus taking significantly less memory than maintaining three separate 34B, 40B, and 80B models. During training, MatFormer samples and optimizes the 34B submodel for X steps, the 40B submodel for Y steps, and the entire 80B model for Z steps. This approach consumes the same total FLOPs as training the individual 34B, 40B, and 80B models separately. The advantages of MatFormer include:
>
> Enhanced Model Quality: MatFormer submodels, particularly at smaller scales, benefit from extended training of shared parameters, resulting in higher quality compared to baseline models of the same size (Appendix B4, 1st point).
>
> Mix’n’Match: MatFormer provides 1000s of models through Mix’n’Match without requiring additional training. In comparison, baseline models only provide the explicitly trained 34B, 40B, and 80B models. Post-hoc techniques such as pruning or distillation adds substantial computational overhead, as discussed previously.
>
> Thus, MatFormer leverages the same amount of FLOPs as training individual models but achieves greater efficiency and flexibility by consolidating the training process, avoiding the need for extensive post-hoc adjustments.
>
> We hope this explanation clarifies the computational cost and efficiency benefits of using MatFormer relative to training smaller models from scratch.
>
> 7. "... limited to models with less than a billion parameters..." "...worried that using such small models..."
>
> We share your interest in scaling up MatFormer. Although we show this for 850M parameter models due to resource constraints, using MatFormer becomes critical as we scale to larger sizes, where training models with intermediate sizes becomes increasingly challenging. We use these larger models as motivating examples for elastic inference algorithms that enable users to obtain any sized model during inference without any post-training. We hope this clarifies our intention behind mentioning Llama models.
>
> We will try to include billions parameter sized models in the final draft. Additionally, our scaling laws show that MatFormer scales as well as baselines. This suggests that larger models will behave similarly
>
> ---
> We would be very happy to discuss any further questions about the work, and would really appreciate an appropriate increase in score if reviewers’ concerns are adequately addressed.

---

> > ### Comment · Reviewer_6C9L · 2024-08-07
> > **Responses to rebuttal**
> >
> > Thanks for the rebuttal. I would like to continue the conversation:
> > 1 & 3 & 6. The authors are side-stepping the fact that smaller models converge faster. For instance a smaller models would consume fewer tokens until its accuracy saturates. As such, training small models is much cheaper than training large models. On the other hand, how is convergence speed impacted with elastic training? Since you're training many-in-one, do you need to consume more tokens until the accuracy of the many-in-one models saturate?
> >
> > 4. OK and thanks for acknowledging that you only have preliminary results for the attention part.
> >
> > 5. Sounds good. Note that Flextron has experiments for Billion-scale models, as opposed to Matformer's results using Million-scale models. Also Flextron uses an auto-router. But again, because this work is contemporary, I do not take into consideration this in my recommendation.
> >
> > 7. Sounds good, thank you for agreeing that the results need to be scaled up. Good luck trying to add those to the final draft.
> >
> > In conclusion, I appreciate the responses, and I am glad the authors mostly agree with my initial assessment. I maintain that this is a borderline paper leaning on the accept side. But I certainly do not think that the contribution warrants a very strong accept.

---

> > > ### Author Response · Authors · 2024-08-09
> > > **Response to the follow up**
> > >
> > > We thank the reviewer for the prompt reply. We answer their follow up question here:
> > >
> > > > For instance a smaller models would consume fewer tokens until its accuracy saturates ... do you need to consume more tokens until the accuracy of the many-in-one models saturate?
> > >
> > > We do not require more tokens during elastic training, regardless of whether the baseline models to compare against are small or large, rather _**less**_ tokens compared to baselines to reach the same quality and saturation as them. Let’s consider two baseline models, A and B, with model A being smaller or equal in size to model B. Our MatFormer model, MatFormer-B, is of the same size as the larger model B. Within MatFormer-B, consider a subnetwork, MatFormer-A, which mirrors the size and architecture of Baseline-A.
> > > During training, we optimize MatFormer-A for the same number of tokens and steps as Baseline-A and MatFormer-B for the same number of tokens and steps as Baseline-B. Consequently, each granularity in MatFormer is trained for the same number of steps and tokens as its corresponding Baseline model. And the total compute (FLOPs) and total memory required is also same or less than training the two baseline models (Baseline-B, Baseline-A).
> > >
> > > Due to parameter sharing within MatFormer, optimizing either of the granularity MatFormer-A or MatFormer-B also partially trains the other granularity for free in addition, without requiring additional FLOPs. This happens due to parameter sharing, and results in MatFormer-A and MatFormer-B achieving performance _**at least**_  as that of Baseline-A and Baseline-B (Fig. 2 (a))
> > >
> > > In other words, to achieve saturation and quality levels equivalent to Baseline-A and Baseline-B, MatFormer requires using fewer tokens and less compute compared to the Baselines. While using the same compute and tokens results in better performance.
> > >
> > > Note this is not the main advantage of MatFormer. The primary advantage of MatFormer lies in its ability to provide thousands of intermediate models between MatFormer-A and MatFormer-B, all while using the same total FLOPs, tokens, and compute as the Baseline models, which only provide two fixed-size models.
> > >
> > > ----
> > > We welcome further questions about the work, and would really appreciate an appropriate increase in score if reviewers’ concerns are adequately addressed

---

> > > > ### Comment · Reviewer_6C9L · 2024-08-09
> > > > **Question is being side-stepped**
> > > >
> > > > What I am curious about is a convergence curve showing after how many tokens consumed does the small model's loss flatten, and compare that to the many-in-one. Fig 2 (b) is showing something completely unrelated (zero-shot accuracies). Please respect the reviewer's time and don't make them re-check data that was not asked for.
> > > >
> > > > Anyway, the authors should take this as something to study in the future and improve their work. I maintain my score of 5. Good job.

---

### Official Review · Reviewer_vEVM · 2024-07-11

**Soundness:** 3
**Presentation:** 3
**Contribution:** 2
**Rating:** 7
**Confidence:** 4

**Summary:**

The paper introduces Matformer, an elastic modeling strategy for inference that provides flexibility for latency and cost requirements. The authors base their method on the recently proposed matryoshka representation learning (MRL) paradigm to introduce nested substructures in the transformer blocks. Specifically, they introduce this in the FFN layers of the transformer and allocate a hyperparameter $g$ to design the number of granular blocks. In the paper, they use $g=4$, where the granularities are mapped as $\{d_{ff}, d_{ff}/2, d_{ff}/4, d_{ff}/8 \}$, where $d_{ff}$ represents the inner dimension of the FFN blocks. The models are trained following the original loss functions used to train baseline models (i.e., models without MRL blocks). The paper showcases results for decoder-only LLMs, encoder-only ViTs, and adaptive retrieval following a simple NN retrieval metric. The decoder-only LLMs also show that the models exhibit scaling laws similar to the baseline models (with slightly shifted constants) and show how a single model can be used efficiently for speculative decoding scenarios, with an optional attention-cache sharing mechanism.

**Strengths:**

1. The paper presents a simple yet effective method to introduce elasticity into existing transformer architectures without relying on compute-intensive methods such as NAS (though the method is composable with NAS).
2. To simplify the model design, The model relies on the efficient MRL formulation (ie, using nested sub-structures) over $g$ different layers per FFN block.
3. For inference, instead of relying on NAS, the authors propose a simple Mix-n-Match strategy to combine all the different nested sub-substructures. They propose a simple "least slope" increasing method for the granularity of the model for best performance (simplifying the design space for the users)
3. The proposed method applies to different transformer networks (both decoder-only and encoder-only). The authors train decoder-only LLMs and encoder-only ViTs to showcase the method's flexibility.
4. The authors back their paper with comprehensive evaluations, ablation studies, results on scaling trends, and comparisons against relevant baselines (traditional training and other elastic inference methods).
5. For the LLM inference use case, the authors showcase how additional capabilities, such as speculative decoding, are inherent to the model's design.

**Weaknesses:**

1. Scaling the proposed Matformer models will be difficult. For a given model scale, to keep the models "equivalent" to the original baselines (for downstream accuracy/log perplexity loss), the MatLM models, for example, are trained with 4x more tokens. Training for such long durations is computationally intractable at model scales that are deployed today.
    - The scaling trends (from the trained models) also do not look too promising. For example, at the XL scale for MatLM or the B-16 scale for ViTs, the MatLM models seem to perform very similarly to the baseline models. Also, if you look at the loss/accuracy for the MatLM-XL models, you get very similar losses/accuracy to the base model - which "sees" 4x fewer tokens, so overall gains are minimal.
3. While the authors show through ablations that their "least slope" method of Mix-n-Match works best, in practice, it seems difficult to select granularities unless some quick experiments are not done. For e.g., at the model depths presented in the paper, it is to figure out to use M+L blocks to outperform an L baseline potentially, but as depth increases, will relying on only two granularities work - or will there be a need to define a fine-grained slope (still following the least-slope method).
4. While the speculative decoding scenario shown is interesting, generally for effective speedups (1.5x ~ 2.5x) from speculative decoding rely on much smaller draft models relative to target models (for e.g., the original paper referred to in the paper - there is ~10x gap in model sizes). Even if the MatLM models are scaled up, with the current granularities, the smallest model size will be 50%-60% smaller than the largest model size).
    - This may not pave the way to more effective speedups unless more granular blocks $g$ are introduced. However, that is counter-intuitive to the training FLOPs spent as you'd effectively scale up more tokens $\propto$ number of $g$ blocks.
    - Is the intuition here to rely on better consistency and attention-cache sharing capabilities to scale for more effective speedups?

**Questions:**

1. Can the authors elaborate more on the attention-cache sharing mechanism? Is the sharing speed because the draft models see a better cache from the largest model's inference pass (because the largest model has to generate 1 final token) during the subsequent generation phase? And does that eventually help improve the consistency of generated tokens? If so, this favors setups where more tokens are speculated and verified. Can the authors comment on how many tokens they speculate per step?

2. In some places, it is a little difficult to understand what the results correlate to in graphs. For eg, in Section 4.2.2, the authors state "For example, with a loss of < 0.5% accuracy, MatViT-L/16 can reduce compute cost by 40%" - but it is unclear where in the graph this result is evident. There are many other instances like this in the paper. Can the authors either point to such results in the graphs? Or be more verbose in the Appendix about how these results are inferred for the readers to understand.

3. For some of the results, where the authors claim better results - the "betterness" claims are very weak. For e.g., the authors claim 0.35% better accuracy for the L/16 ViT models, but this also seems very close to the seed range for some models of this size. Similarly, for the MatLM results in Section 4.1 - they claim 0.01 log perplexity is a better result than the DynaBERT approach. This is, again, a very minor difference. Can the authors expand on this by showing either seed numbers (which might be difficult in rebuttal time - I understand this) or highlighting other papers that talk about this to understand the significance of the results?

4. For many of the downstream results in the Appendix, several results in Tables 9 and 10 are very close to random chance accuracy - making it a little difficult to gauge the gains from using the MatLM training recipe. Can the authors comment on this? Are some tasks more representative of the model's performance at these scales, which can help readers understand the significance of the results?

**Limitations:**

Yes, the limitations of most results and methods are presented throughout the paper. Nothing extra is needed.

---

> ### Author Rebuttal · Authors · 2024-08-06
>
> We thank the reviewer for their thoughtful comments and support. We clarify the main concerns below:
>
> 1. "Scaling ... will be difficult...Training .. intractable at model scales that are deployed today."
>
> We clarify that MatFormer does not use more data or compute compared to the baselines trained separately and can be scaled similarly to the baselines. Rather than training different sized baseline models from scratch and separately, our approach trains them together and thereby enjoys the benefits of training the common parameters jointly.
>
> Consider 4 baselines trained using FLOPs-S/M/L/XL for Baseline-S/M/L/XL, resulting in total FLOPs_TOTAL=FLOPs_S+...+FLOPs_XL. These separate baseline models, collectively use memory equivalent to Memory-S+Memory-M+Memory-L+Memory-XL.
> MatFormer, in comparison, has memory=max(Memory-S,...,Memory-XL)=Memory-XL. The FLOPs used by MatFormer does not exceed FLOPs-TOTAL.
>
> 2. "The scaling trends... do not look too promising..."
>
> The aim of MatFormer is to provide 1000s of models along a #param-vs-performance curve that is at least as good as the baseline #param-vs-performance curve, while using less memory and the same FLOPs as training a fixed number of baselines. Note that MatFormer gives 1000s of models on this optimal curve for free, whereas the baseline provides only a fixed number of explicitly trained models (4 in our paper). MatFormer achieves this (Figure 2), where its #param-vs-performance curve matches around the XL scale but outperforms the baseline significantly at smaller scales.
>
> Additionally, it is possible to adjust the sampling probability while maintaining the same total FLOPs, thereby achieving higher performance across models from S to XL (as shown in Table 6). We will add this discussion to the paper to clarify this.
>
> 3. "While the authors show through ablations their "least slope" method of Mix-n-Match..."
>
> In initial experiments, we found that this trend was consistent at different model scales, and the least slope method can be calculated mathematically for a given latency budget. The intuition behind this is that since "uniform" submodels are used during training, i.e. all layers are S/M/L/XL, making the least change to these explicitly trained granularities ought to produce the best results. We further discuss this in Appendix D1, and believe same heuristic should work when generalizing MatFormer further.
>
> 4. "While the speculative decoding...effective speedups unless more granular blocks are introduced."
>
> Ans: Thank you for the insightful comment. We believe that the smallest model can be much smaller than 50% of the XL model, especially as we scale to much larger sizes. Consider the Llama-3.1 70B model, with model_dim=8k. We can have the smallest model as 1/16th of the total parameters, resulting in the S model having approximately ((70-2.5)/16+2.5)≈7B parameters. This is ~10x smaller than the 70B XL model.
>
> Yes, having more granularities would allow for a much smaller S model. To avoid higher training costs, the smallest granularity can be quite small while maintaining the same n(granularities) - we could have 1/16, 1/4, 1/2 as granularity ratios for the S, M, L models relative to the XL model. Thank you for highlighting this, we will incorporate this discussion into the paper.
>
> 5. Is the intuition here to rely on better consistency ... more effective speedups?
>
> Yes, better consistency will contribute to more effective speedups. We presented significantly higher consistency as a key advantage of MatFormer and illustrated two of the many possible use cases for it. For Decoder LMs, speculative decoding benefits from this consistency (Section 4.1.1). For Encoders, it enables adaptive retrieval (Section 4.2.2).
>
> 6. "Can the authors elaborate more on the attention-cache sharing..."
>
> You are correct. Since S,XL models share attention blocks, when the XL model verifies the S model’s drafts, it performs a forward pass, resulting in the generation of the XL model’s KV Cache. This cache can overwrite the S model’s cache. This is not possible with baselines because their embedding spaces are different. We verified this empirically and found that baselines failed completely.
>
> We observed higher speedups when sharing the attention-cache (Table 2). For our experiments we speculated 3 draft tokens before verification through the larger model.
>
> 7. "...little difficult to understand what the results correlate to in graphs..."
>
> Thank you for pointing this out. In the MatViT instance, we were referring to Fig. 5(b). Here, the Mix’n’Match model having 175M parameters has less than 0.5% accuracy drop compared to the XL model which has 300M parameters, resulting in ~40% speedup. We’ll clarify this and go over the paper to ensure there are no ambiguities.
>
> 8. "... seed numbers... the significance of the results?"
>
> For ImageNet-ViT models, the variance between runs is marginal – which we observed across multiple seeds. 0.35% gain in the range of 85% accuracy is statistically significant. We also want to mention that this gain is a bonus and is not the main takeaway for MatFormer. For LMs, 0.01 log perplexity is quite significant. On two different seeds we gave ~15% more FLOPs to DynaBERT than MatFormer, but the 0.01 log perplexity gap was not bridged. We’ll clarify this further in the final draft.
>
> 9."... readers understand the significance of the results?"
>
> At small scales, the models may not be sufficient to provide meaningful comparisons for individual evaluation tasks, which can result in noise. To mitigate this, we use 25 different evaluation tasks and average the performance over them to judge the final performance. This provides a more reliable assessment of the model's capabilities, which strongly correlate with loss/perplexity. In future work, we aim to scale to even larger models.
>
> ---
>
> We welcome further questions about the work, and would really appreciate an appropriate increase in score if reviewers’ concerns are adequately addressed.

---

> > ### Comment · Reviewer_vEVM · 2024-08-08
> >
> > Thank you for the detailed response to the reviews and additional experimental results. After reading all reviews and response, and the overall global response, I will update my score to accept (score: 7).
> >
> > Please incorporate necessary changes for the unclear sections in the revised draft.

---

### Official Review · Reviewer_zcbm · 2024-07-13

**Soundness:** 4
**Presentation:** 4
**Contribution:** 4
**Rating:** 9
**Confidence:** 4

**Summary:**

This paper presents MatFormer, a nested Transformer architecture for elastic inference deployment constraints. It follows the principle of matryoshka representation learning and incorporate nested structure in the FFN modules of Transformers. Experiments show that MatFormer can (1) reliably obtain 582-850M model from a single 850M model, (2) preserve the metric-space structure for adaptive large-scale retrieval for extracted encoder, (3) friendly to speculative decoding.

**Strengths:**

(1) Important problem and a very interesting solution.

(2) The writing is straightforward and easy to understand.

(3) Results are comprehensive and multi-dimensional. In additional to the main claim, its ability to be reliably used in speculative decoding shows the generalization ability of the methodology

**Weaknesses:**

The reviewer does not think there are major problems. Please see questions.

**Questions:**

(1) Can you add a discussion section with MoE?
(2) Is it possible to obtain a MatFormer model directly from a pretrained checkpoint (e.g. in some form of up-cycling)?

**Limitations:**

Please see question 2.

---

> ### Author Rebuttal · Authors · 2024-08-05
>
> We thank the reviewer for their appreciation of our contributions, and answer the questions asked below:
>
> --------
>
> 1. Can you add a discussion section with MoE?
>
> **Answer**: We agree that a discussion section on MoE would be appropriate, and will include a discussion section on this in the final draft. Both Matformer and MoE models are conditional computation architectures that can activate certain portions of the model depending on the input. For dynamic workloads, where the compute resources or the input hardness changes for each model query, we can use the universal MatFormer model to dynamically extract the optimal submodel for token-based routing in LLMs, similar to MoE models that focus on inference efficiency (Kudugunta et al., 2021; Li et al., 2022).
>
> - Kudugunta, S., Huang, Y., Bapna, A., Krikun, M., Lepikhin, D., Luong, M. T., & Firat, O. (2021). Beyond distillation: Task-level mixture-of-experts for efficient inference.
> - Li, M., Gururangan, S., Dettmers, T., Lewis, M., Althoff, T., Smith, N. A., & Zettlemoyer, L. (2022). Branch-train-merge: Embarrassingly parallel training of expert language models.
>
> ---------
>
> 2. Is it possible to obtain a MatFormer model directly from a pretrained checkpoint (e.g. in some form of up-cycling)?
>
> **Answer**: It is indeed possible to obtain a MatFormer model directly from a pretrained checkpoint. In Table 7 (Appendix G1), we validate this with MatViT. While training MatViT from scratch results in more accurate submodels, we are still able to obtain deployable submodels by finetuning ViT with the MatViT objective using 2% of the training budget.

---

> > ### Comment · Reviewer_zcbm · 2024-08-10
> >
> > Thank you! This is great. I am particularly interested in the idea of up-cycling for a potential future work - I will keep the original score of 9. Please consider accept this paper!

---

> > > ### Author Response · Authors · 2024-08-10
> > >
> > > Thank you for the support of the work. We are excited about the future applications of MatFormer as well!

---

### Official Review · Reviewer_51tp · 2024-07-19

**Soundness:** 3
**Presentation:** 3
**Contribution:** 3
**Rating:** 6
**Confidence:** 3

**Summary:**

The authors proposed a novel Transformer architecture called MatFormer to provide elastic inference across diverse deployment constraints. Specifically, the authors incorporate a nested Feed Forward Network (FFN) block structure within a standard Transformer model. During training, the authors optimize the parameters of multiple nested FFN blocks with varying sizes, enabling the extraction of hundreds of accurate smaller models without incurring additional computational costs. Experimental results on different model classes (decoders and encoders) and modalities (language and vision) demonstrate the effectiveness of the proposed method. My detailed comments are as follows.

**Strengths:**

1. The idea of incorporating a nested sub-structure within the standard Transformer and optimizing all the g granularities to produce a single, universal elastic model is interesting.

2. The paper introduces Mix’n’Match, a simple heuristic with no computation overhead that finds optimal submodels within a given parameter budget, outperforming more complex NAS methods without any training cost.

3. The results show the proposed method generalizes well to both decoder-only language models (MatLM) and vision encoders (MatViT), which has great potential in practice.

4. The paper is easy to read and provides enough experimental details to reproduce.

**Weaknesses:**

1. The idea of jointly optimizing a nested sub-structure is similar to Slimmable networks [A] and the supernet in Neural Architecture Search. More explanations are required to clarify the differences between them. What are the new challenges of applying the nested sub-structure to transformer architectures?

2. Some important details of dynamic workloads are missing. It would be better for the authors to show more details about how to use the universal MatFormer model to dynamically extract the optimal submodel for each token or query.

3. Although the paper shows promising results in experimental settings, it lacks extensive evaluation in real-world deployment scenarios. It would be better for the author to deploy the quantized models on GPUs or CPUs and report the memory consumption, inference speed as well as accuracy.

[A] Slimmable neural networks. ICLR 2019.

**Questions:**

NA

---

> ### Author Rebuttal · Authors · 2024-08-05
>
> We thank the reviewer for their thoughtful comments and support towards the paper. We clarify the main concerns raised by the reviewer:
>
> -----------
>
> 1. The idea of jointly optimizing a nested sub-structure is similar to Slimmable networks [A] and the supernet in Neural Architecture Search. More explanations are required to clarify the differences between them. What are the new challenges of applying the nested sub-structure to transformer architectures?
>
> **Answer**: Slimmable networks optimizes all models simultaneously. This idea is also used in DynaBERT, where it applies this training technique on transformer based architecture rather than on CNNs like Slimmable Networks. Hence, as a baseline we compare with DynaBERT (which is also more recent) and demonstrate the advantages of the MatFormer training methodology, which optimizes only one model at a time. This results in more gradient updates for the same FLOPs and memory. Furthermore, neither Slimmable Networks nor DynaBERT incorporate a strategy for selecting submodels beyond the few explicitly trained submodels. In contrast, MatFormer employs Mix’n’Match as an efficient and near-optimal model selection strategy, offering 1000+ models at no additional cost. We will clarify these differences further in the revised paper draft.
>
> -----------
>
> 2. Some important details of dynamic workloads are missing. It would be better for the authors to show more details about how to use the universal MatFormer model to dynamically extract the optimal submodel for each token or query.
>
> **Answer**: In this paper, we primarily focus on pretraining a single model that can result in multiple performant submodels without any additional training. It is possible to use these resulting submodels with many types of algorithms that are geared towards using multiple models to improve latency such as speculative decoding (Leviathan et al., 2023; Kim et al,. 2023), model routing (Ong et al., 2024; FrugalGPT) and cascade algorithms geared towards inference efficiency  (Narasimhan et al 2024, Kolawole et al 2024).
>
> There are recent works which build on top of our work and apply routing for dynamic workloads. We skip mentioning them to not break double blind policy, but will certainly include a discussion in our revised version. We believe that developing new algorithm variants to use MatFormer for dynamic workloads is a promising area for future work, and will add a discussion of this to the final draft.
>
> - Y. Leviathan, M. Kalman, and Y. Matias. Fast inference from transformers via speculative 479 decoding. 2023.
> - Kim, Sehoon, Karttikeya Mangalam, Suhong Moon, Jitendra Malik, Michael W. Mahoney, Amir Gholami, and Kurt Keutzer. "Speculative decoding with big little decoder." Advances in Neural Information Processing Systems 36 (2024).
> - Ong, I., Almahairi, A., Wu, V., Chiang, W. L., Wu, T., Gonzalez, J. E., ... & Stoica, I. (2024). RouteLLM: Learning to Route LLMs with Preference Data. arXiv preprint arXiv:2406.18665.
> - Chen, L., Zaharia, M., & Zou, J. (2023). Frugalgpt: How to use large language models while reducing cost and improving performance. arXiv preprint arXiv:2305.05176.
> - Narasimhan, H., Jitkrittum, W., Rawat, A. S., Kim, S., Gupta, N., Menon, A. K., & Kumar, S. (2024). Faster Cascades via Speculative Decoding. arXiv preprint arXiv:2405.19261.
> - Kolawole, S., Dennis, D., Talwalkar, A., & Smith, V. (2024). Revisiting Cascaded Ensembles for Efficient Inference. arXiv preprint arXiv:2407.02348.
>
> -----------
>
> 3. Although the paper shows promising results in experimental settings, it lacks extensive evaluation in real-world deployment scenarios. It would be better for the author to deploy the quantized models on GPUs or CPUs and report the memory consumption, inference speed as well as accuracy.
>
> **Answer**: We thank the reviewer for their feedback, and will include these details in the final draft.
>
> We welcome further questions about the work, and if key issues are addressed, we would greatly appreciate an appropriate increase in score.

---

> > ### Comment · Reviewer_51tp · 2024-08-12
> > **Official Comment by Reviewer 51tp**
> >
> > Thanks for your detailed responses. After reading all reviews and responses, I will maintain my score to accept.

---

### Decision · Program_Chairs · 2024-09-25

**Decision:**

Accept (poster)

**Comment:**

This submission presents a novel Transformer architecture called MatFormer to provide elastic inference across diverse deployment constraints. Specifically, the authors incorporate a nested Feed Forward Network (FFN) block structure within a standard Transformer model. During training, the authors optimize the parameters of multiple nested FFN blocks with varying sizes, enabling the extraction of hundreds of accurate smaller models without incurring additional computational costs. The authors should consider the reviewers' comments and amend the issues raised by reviewers.